# High-throughput in vivo mapping of RNA accessible interfaces to identify functional sRNA binding sites

Mia K. Mihailovic[1], Jorge Vazquez-Anderson[1], Yan Li[2], Victoria Fry[1], Praveen Vimalathas[1], Daniel Herrera[3], Richard A. Lease[4,5], Warren B. Powell[2] & Lydia M. Contreras[1]

Herein we introduce a high-throughput method, INTERFACE, to reveal the capacity of contiguous RNA nucleotides to establish in vivo intermolecular RNA interactions for the purpose of functional characterization of intracellular RNA. INTERFACE enables simultaneous accessibility interrogation of an unlimited number of regions by coupling regional hybridization detection to transcription elongation outputs measurable by RNA-seq. We profile over 900 RNA interfaces in 71 validated, but largely mechanistically under-characterized, *Escherichia coli* sRNAs in the presence and absence of a global regulator, Hfq, and find that two-thirds of tested sRNAs feature Hfq-dependent regions. Further, we identify in vivo hybridization patterns that hallmark functional regions to uncover mRNA targets. In this way, we biochemically validate 25 mRNA targets, many of which are not captured by typically tested, top-ranked computational predictions. We additionally discover direct mRNA binding activity within the GlmY terminator, highlighting the information value of high-throughput RNA accessibility data.

[1] McKetta Department of Chemical Engineering, University of Texas at Austin, 200 E. Dean Keeton St., Stop C0400, Austin, TX 78712, USA. [2] Department of Operations Research and Financial Engineering, Princeton University, Sherrerd Hall, Charlton St., Princeton, NJ 08544, USA. [3] Department of Computer Science, University of Texas at Austin, 2317 Speedway Stop D9500, Austin, TX 78712, USA. [4] Department of Chemical and Biomolecular Engineering, The Ohio State University, 151W. Woodruff Ave, Columbus, OH 43210, USA. [5] Department of Chemistry and Biochemistry, The Ohio State University, 100W. 18th Ave, Columbus, OH 43210, USA. These authors contributed equally: Mia K. Mihailovic, Jorge Vazquez-Anderson. Correspondence and requests for materials should be addressed to L.M.C. (email: lcontrer@che.utexas.edu)

Bacterial small regulatory RNAs (sRNAs) constitute a distinctive class of RNAs that possess intrinsic roles in cellular regulation[1,2]. To respond to environmental stress, sRNAs tune metabolic and regulatory pathways, typically by altering messenger RNA (mRNA) translation or stability via direct association[3,4]. To date, a relatively small proportion of confirmed sRNAs (e.g., >100 in *E. coli*[5,6], >70 in *M. tuberculosis*[7], >40 in the non-pathogenic *M. smegmatis*[7,8]) have been mechanistically characterized (e.g., <20 in *E. coli*)[5]. The challenges of functional sRNA characterization can be partly attributed to specific characteristics such as: regulation of targets in trans, regulation of multiple targets per sRNA, and relatively small regulatory regions[2] (10–25 nt[9] with 8–9 nt[10] of imperfect complementary). sRNA interactions are initiated via seed sequences of pairing regions that canonically reside in single-stranded (unstructured) sequences or in hairpin apical loops, typically located in the first 2/3 of sRNA sequences[1]. In recent years, computational target prediction programs have incorporated various facets of sRNA regulation, including seed parameters[11], to offer thousands of putative bacterial sRNA:mRNA pairs; however, only a small set has been successfully validated in vivo[2]. This can be attributed to true targets (energetically) ranking well into the hundreds of predicted targets in popular algorithms[12] coupled with the demanding nature of traditional partner validation assays (i.e., each potential sRNA:mRNA interaction is tested one at a time).

Another confounding factor hindering sRNA characterization is the regulatory impact of the Hfq chaperone, an Sm-protein known to affect sRNA:mRNA interactions. The extent to which Hfq globally impacts the ability of all sRNAs to establish interactions with cognate mRNA targets is a fundamental question concerning intracellular proteins' effects on sRNA networks. Because its intracellular quantity is limited, sRNAs are in constant competition for Hfq and are likely cycled off of Hfq once they have established stable base pairing with an mRNA target[13,14]. Many mechanistic details of the Hfq-binding RNA sequence have been well-studied, uncovering characteristic binding motifs and interaction mechanisms, such as its ability to recognize a number of sRNA Rho-independent terminators[13,15]. However, the mechanistic necessity of Hfq for proper RNA interactions remains elusive due to a combination of factors, such as under-realized insights into the hierarchy of Hfq-sRNA regulation[15], cross-regulation of sRNAs by other putative chaperones, such as ProQ[16], absence or limited function of Hfq homologues within GC-rich Gram-positive bacteria[17], and variable roles of Hfq on different sRNA:mRNA pairs[14,15]. Thus far, it has been proposed that sRNAs rely on Hfq to increase local concentrations of sRNA and respective target mRNA[18,19], stabilize interaction partners, facilitate base pairing catalysis by structural rearrangement[13,15] or present some combination thereof, depending on the target mRNA[3].

In recognition of the need for high-throughput mechanistic sRNA insights in vivo, recent efforts have unveiled large sets of RNA:protein and RNA:RNA interactions and even corresponding interacting regions[20–24]. However, these methods are somewhat limited by requisite targeting toward a single protein binding partner of interest[24,25], as well as by difficulties sensing interactions involving low-abundance non-coding RNAs[23]. To this end, efforts have recently been made to complement computational predictions with in vivo insights to enhance prediction reliability. For example, in vivo chemical and enzymatic probing methods gauge the level of "protection" or reactivity of individual bases/ backbones within a region of interest, and have recently been adapted for high-throughput use[26,27]. As these local nucleotide availabilities do not always correlate with regional-level accessibility that more accurately mimics RNA:RNA interactions of regulatory interfaces[28], efforts have also been placed on methods

to quantify regional in vivo RNA hybridization. Corresponding datasets have yielded useful predictions of regions capable of establishing RNA:RNA interactions[28]. Nonetheless, acquisition of these data is typically limited by low-throughput experiments[28,29] and reliance on sRNA overexpression that disturbs native transcript or protein stoichiometry[30]. As such, there remains a need for high-throughput methods to identify characteristic features of RNA regions that are likely to engage in regulatory interactions with respective mRNA targets.

To obtain molecular insight concerning global RNA function in vivo, we developed INTERFACE, **in** vivo transcriptional **e**longation analyzed by **R**NA-seq for **f**unctional **a**ccessibility **c**haracterization in a single **e**xperiment. INTERFACE is a high-throughput method capable of surveying the regional accessibility of a large collection of RNAs simultaneously by mimicking in vivo antisense hybridization. In vivo antisense hybridization has previously been shown to be sensitive to sites involved in intermolecular interactions and to capture transient configurations often relevant to regulation[28,31]. INTERFACE is an engineered RNA system that exploits conserved bacterial mechanisms of translational stalling and Rho-dependent transcription termination mechanisms to quantify RNA hybridization via a transcriptional elongation response. In this work, we apply machine learning to design over 900 antisense oligonucleotide probes to interrogate the accessibility of regions fully covering a library of 71 experimentally validated sRNAs in *E. coli* in their native environment. These sRNAs represent two distinct subclasses: mechanistically characterized sRNAs, meaning that exact binding sites corresponding to at least one mRNA target have been confirmed (27 total sRNAs, of which 16 are used as a training set and the remaining 11 are probed blindly), and mechanistically uncharacterized sRNAs (44 total) (Supplementary Data 1). To demonstrate the ability of INTERFACE to capture global impacts of relevant intracellular factors on sRNA accessibility landscapes, we assess sRNA accessibility profiles both in wild-type *E. coli* BW25113 and in an isogenic strain in which the well-characterized Hfq chaperone is knocked out[32]. In this way, we propose hybridization dependency or independency for 14 sRNAs whose relationship with Hfq is not characterized in the literature. We also identify a global pattern of either "extreme" low or high likelihood for antisense RNA (asRNA) hybridization in regions harboring mapped mRNA-binding sites in 16 well-characterized sRNAs that comprise our training set. This information enables identification of likely functional sites in other sRNAs to serve as "in vivo filters" of computational predictions[11] to identify true mRNA targets in vivo. Upon benchmarking the value of these "INTERFACE-informed" computational predictions against typically tested, top-ranked predictions via in vitro binding assays, we confirm thirteen novel mRNA targets for three uncharacterized sRNAs (SroE, SroG, Tpke70). Of these validated targets, six correspond exclusively to INTERFACE-informed computational predictions, one exclusively to top-ranked computational predictions, and six are shared by both approaches. We additionally confirm twelve INTERFACE-informed targets in six sRNAs that represent both uncharacterized and characterized sRNAs (SroH/ Tpke11/SroA, and CyaR/GcvB/GlmY, respectively), of which half fall below the top-20 in computation prediction ranking. Finally, we showcase a trans antisense regulatory potential within the GlmY terminator as suggested by INTERFACE accessibility data; to our knowledge, GlmY has only been implicated in indirect mRNA regulation via RapZ protein sequestration[33].

## Results

### Enabling high-throughput RNA accessibility characterization.
Assessment of hybridization potential has proven informative in identifying RNA interfaces accessible to RNA:RNA interactions

but has previously been limited to low-throughput approaches[28,31]. To allow large-scale identification of accessible RNA interfaces via next generation sequencing (NGS), we constructed a system that harnesses transcriptional regulation by coupling in vivo asRNA:RNA hybridization to transcription elongation control. In short, this approach allows sensing of molecular interactions between an asRNA toehold-switch "probe" (encoded by a plasmid-based expression system) and a complementary region in a target RNA (taRNA) by virtue of hybridization-activated transcription anti-termination (described below). The anti-termination output was inspired by *trp* operon attenuation[34] that was previously used to synthetically convert translational to transcriptional control in bacteria[35]. In this engineered system, differential transcriptional elongation, quantified via RNA-seq, is a direct function of the propensity for a taRNA region to interact with a corresponding asRNA probe. Short and long transcripts indicate low and high asRNA probe hybridization in a targeted region, respectively.

The inducible system is composed of five main elements that together enable transcription elongation in response to asRNA probe:taRNA hybridization (Fig. 1a). (i) The asRNA probe is a variable length sequence (9–26 nt) complementary to a region of interest within the taRNA. Downstream of the asRNA probe is (ii) the ribosomal binding site sequestration element (RSE) and (iii) a strong ribosomal binding site (RBS). These three elements can form a structure reminiscent of a toehold-switched hairpin loop[36], in which the asRNA probe is the toehold and the RSE and RBS form the hairpin. This is followed by (iv) the elongation switch (ES), which consists of a *tnaC* (*trp* attenuator) leader peptide nucleotide sequence followed by a Rho utilization (*rut*) transcription termination site. Finally, located directly downstream is (v) the RNA elongation reporter (RER), consisting of a truncated GFP coding sequence that supports maximum possible transcript lengths. If a taRNA region is accessible to the asRNA probe, elongated transcripts are generated. This output is stimulated by successful binding of the probe to its target region, which disrupts the hairpin loop structure and enables translation of the *tnaC* leader peptide preceding the *rut* site (Fig. 1b, left). In the presence of tryptophan, the *tnaC* leader peptide induces nascent polypeptide-mediated ribosome stalling which occludes the *rut*[37]. Due to *rut* occlusion, the Rho protein cannot efficiently terminate transcription and allows extended transcriptional elongation into the RER. Conversely, if asRNA probe hybridization with its target region is not established due to inaccessibility of the taRNA region, truncated, or partial transcripts are produced. This occurs because the toehold hairpin is not switched open and *tnaC* translation is not initiated (Fig. 1b, right); thus, the *rut* site remains available to the Rho factor and routine transcription termination occurs. Because a unique probe sequence (corresponding to a unique taRNA region) is incorporated within each INTERFACE transcript, NGS results not only contain information pertaining to transcript length, indicative of taRNA region accessibility, but also a unique sequence identifier of the corresponding target region. Although this sequence identifier is reminiscent of barcoding approaches that have been used in genome-wide RNA structure probing methods, this system yields specific, "bar-coded" outputs for each interrogated taRNA region, rather than variable RNA-specific fragment outputs[26]. In this way, INTERFACE supports the simultaneous characterization of regional accessibility profiles within any number of RNAs.

As shown in Fig. 1c, the implementation of INTERFACE in RNA characterization consists of seven main steps: (i) plasmid library preparation and generation, (ii) library transformation, (iii) polyclonal culture growth and INTERFACE plasmid induction, (iv) total RNA extraction (v) cDNA library generation, (vi) RNA-seq, and (vii) downstream data analysis. Specifically, a library of asRNA probes—each of which is an 9–26 nt long asRNA oligonucleotide complementary to a specific taRNA region is individually inserted into the desired INTERFACE plasmid (Supplementary Fig. 1). The pool of plasmids is transformed as a library into the relevant bacterial strain (in this case *E. coli*) and the system induced in early log growth. Once a user-specified growth phase is reached, total RNA is extracted from the library culture, and a cDNA library is generated for deep sequence analysis. Importantly, fragmentation of RNA is omitted to preserve the ability to correctly assign a 3′-end to the corresponding 5′-end of each transcript and reliably identify the extent of transcriptional elongation for each transcript (based on the taRNA to which the distinct asRNA probe hybridized). Following cDNA purification, samples are sequenced using a standard paired-end Illumina-platform protocol. Finally, corresponding forward and reverse RNA-seq reads are paired and filtered for INTERFACE transcripts corresponding to distinct asRNA probes; asRNA probe-specific transcriptional elongation can then be correlated with the accessibility of respective taRNA regions.

**Validating molecular features that enable INTERFACE.** Three basic mechanistic premises enable INTERFACE to capture regional accessibility: (1) asRNA probe binding to a corresponding taRNA region disrupts the stem loop to expose the RBS, (2) RBS exposure governs transcriptional elongation of the INTERFACE RNA system, and (3) elongation patterns are dependent on the elongation switch and are reliable indicators of regional accessibility. First, asRNA probe hybridization to its taRNA region has been previously verified to expose the RBS using a low-throughput, fluorescence-based asRNA hybridization assay[31]. Second, we confirmed that RBS exposure governs transcriptional elongation by designing two control INTERFACE constructs to represent the extremes within the spectrum of RBS exposure: (i) a permanently sequestered RBS (RSE stably sequesters RBS due to enhanced complementarity) and (ii) a permanently opened RBS (RSE is mutated to expose the RBS). Both constructs contained identical, random asRNA probes predicted to have limited interaction with the transcriptome. From NGS data, we observed three major transcript length peaks present at different loci of the INTERFACE transcript (Fig. 2a): (i) ~80 nt, terminating within the ES sequence (specifically *tnaC*), (ii) ~140 nt, terminating at the end of the *rut* terminator and (iii) ~200 nt, terminating at an RER position that denotes the full extent of transcriptional elongation. Consistent with expectations, the sequestered RBS and the open RBS controls exhibited partial (terminating at *tnaC*) and full (terminating in RER) transcript lengths, respectively (Fig. 2a).

Third, to test the dependency of differential transcriptional elongation on the elongation switch, we assayed the hybridization capacity of well-characterized regions[31] along the model *Tetrahymena* group I (gI) intron taRNA in vivo using an INTERFACE plasmid that supports taRNA overexpression (O-INTERFACE, Supplementary Fig. 1A). Figure 2b shows representative INTERFACE probing results for two regions within this model RNA, one highly inaccessible and one highly accessible (nucleotides 361-375 and 399-414, respectively), as previously reported using a low-throughput assay[31]. Importantly, we observed the expected shift toward longer transcripts for the taRNA region of the gI intron most accessible to its corresponding asRNA probe, but only in the presence of both the ES and the taRNA (Fig. 2b). Moreover, this bias toward larger transcript sizes was absent from transcriptomic data when probing the highly inaccessible region (Fig. 2b, compare upper and lower right panels).

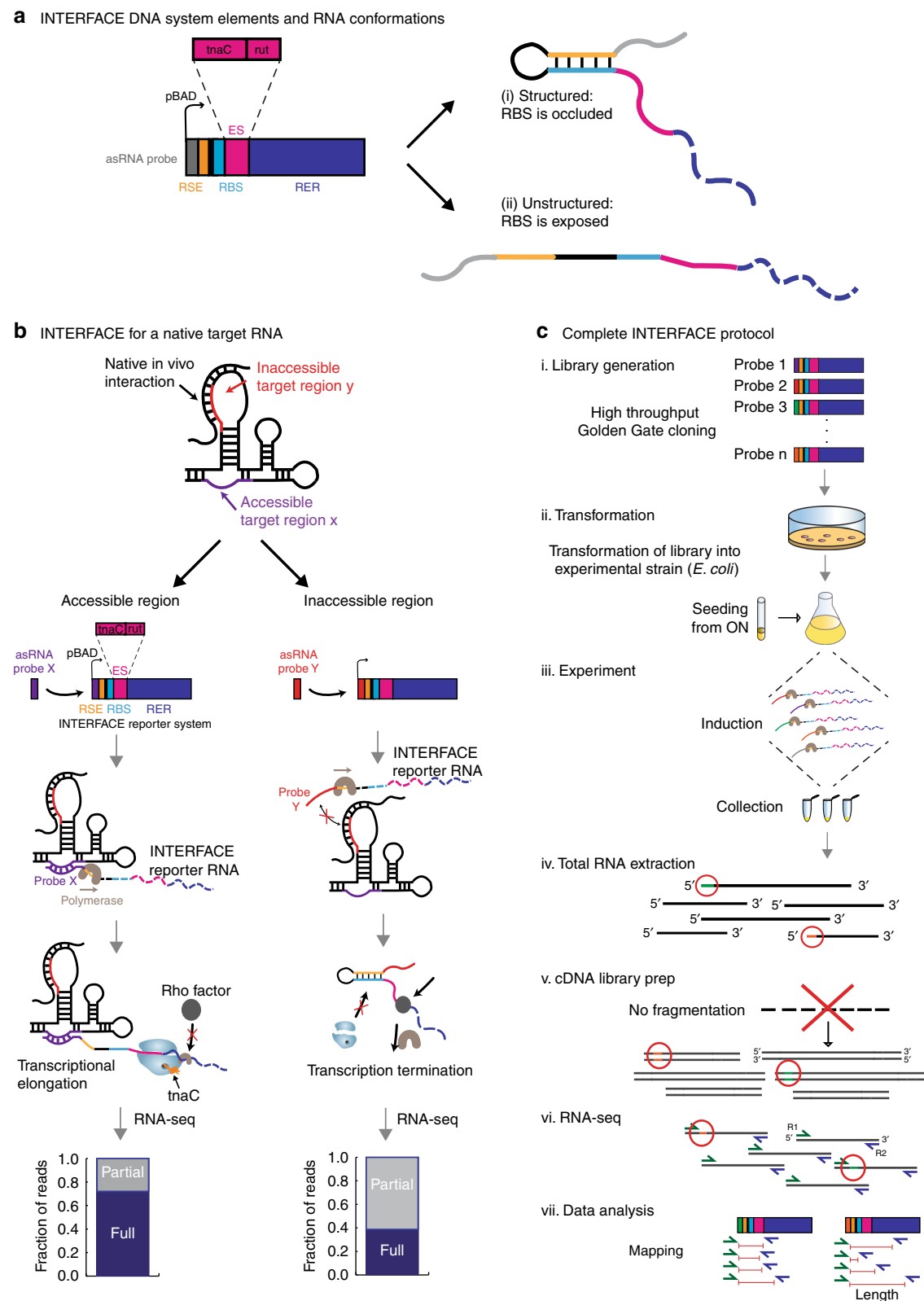

To evaluate the expected high-throughput potential of this approach, we fully characterized the gI intron in a single in vivo experiment via transcriptomics analysis of an O-INTERFACE plasmid library targeting 30 unique regions within the gI intron (in a combination of 10-mer and 16-mer probe sequences for 100% coverage) (Fig. 2c, Supplementary Fig. 2). The regional accessibility results from these 30 asRNA probes are consistent with evidence from previous studies[31], validating the ability of this method to assess hybridization accessibilities in a high-throughput manner via NGS.

**Functional regions of a native sRNA ensemble revealed**. To assess universal molecular features of sRNAs that contribute to

**Fig. 1** Elements, utility, and complete protocol of INTERFACE. **a** Moving from 5′ to 3′, each INTERFACE is composed of a unique asRNA probe, followed by the RSE, then the ES, comprised of *tnaC* and the *rut* site, and then the RER (a truncated *gfp* gene). INTERFACE can adopt two distinct RNA conformations depending on asRNA probe:taRNA region binding that determines the exposure of the RBS upstream of the ES: (i) a structured hairpin in which the RBS is occluded and (ii) an unstructured domain in which the RBS is exposed. **b** When targeting an accessible region (left panel), the INTERFACE asRNA probe 'X' binds strongly, releasing the RBS for *tnaC* translation. Translation of *tnaC* causes ribosomal stalling near the *rut*. The stalled ribosome physically occludes the Rho factor from the *rut* site which enables transcriptional elongation. In contrast, INTERFACE targeting of an inaccessible RNA region with asRNA probe 'Y' (right panel) generates a truncated transcript. Specifically, the absence of *tnaC* expression due to lack of binding between the asRNA probe and the taRNA enables *rut* site availability for Rho-dependent transcriptional termination. **c** (**i**) A library of asRNA probes is inserted into either of two parent plasmids (Supplementary Fig. 1). (**ii**) The library is transformed into the relevant experimental strain as described in Methods section. (**iii**) The INTERFACE system is induced and the library of asRNA probes interact with corresponding taRNA regions in vivo to varying degrees, generating a spectrum of transcript lengths bar-coded with the probe sequence at the 5′-end. (**iv**) Total RNA, including transcripts of interest containing corresponding probes (green and orange 5′-ends circled in red), is extracted. (**v**) A cDNA library for Illumina-platform RNA-seq is generated, in which INTERFACE RNA is among the pool of adapter-ligated RNA and consequently in the final cDNA library (red). (**vi**) RNA-seq is performed per standard procedures in a paired-end 75 × 2 run in the Illumina platform. (**vii**) Mapping against synthetic INTERFACE library is performed to exclude non-INTERFACE transcripts and associate corresponding 3′-ends (R2) to 5′-ends (R1) to determine the length of each region-specific INTERFACE transcript

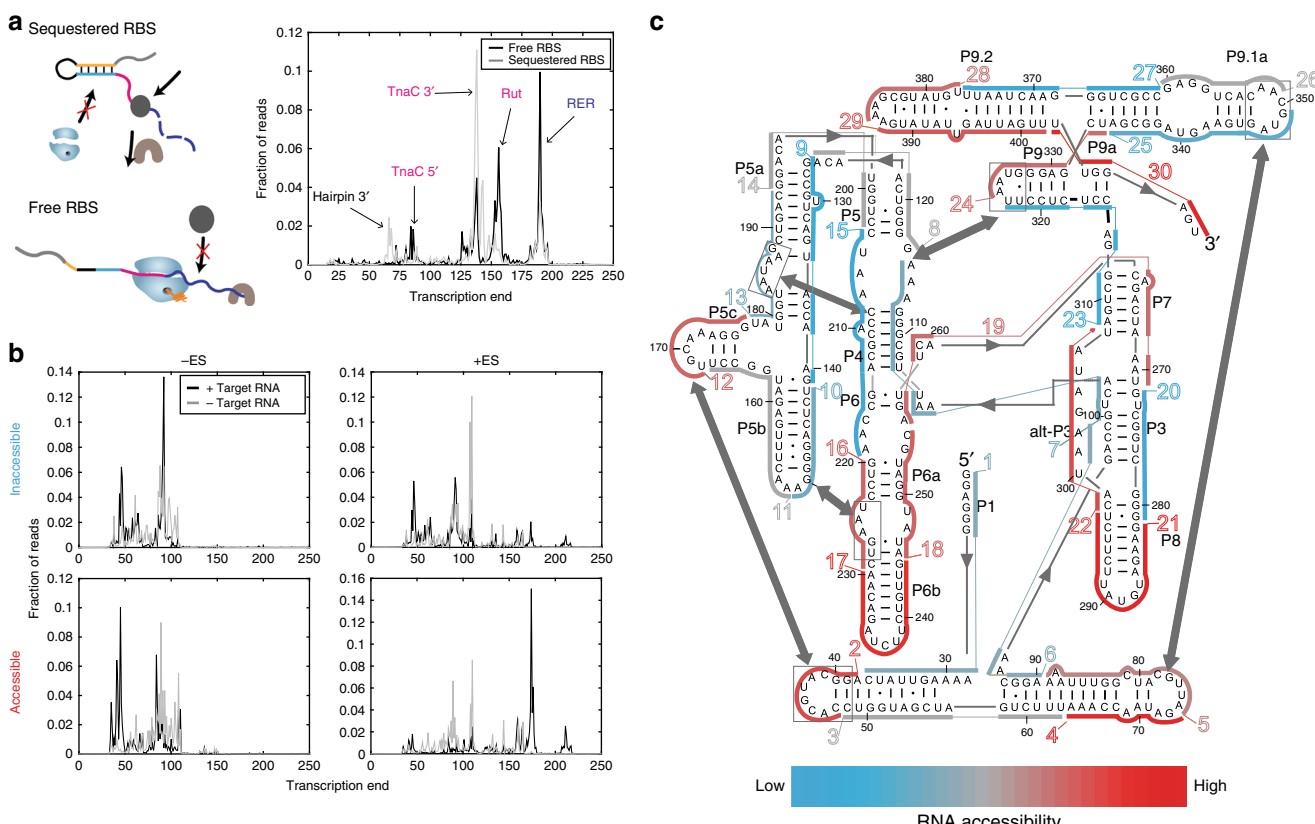

**Fig. 2** INTERFACE supports high-throughput characterization of hybridization landscapes. **a** RBS exposure influences transcriptional elongation. The sequestered RBS control, expected to prevent initiation of *tnaC* translation and allow Rho-dependent transcription termination (top left), shows shorter transcriptional elongation (gray) than the free RBS control (black), expected to allow *tnaC* translation to cause ribosomal stalling and prevent Rho termination (bottom left). Traces depict the weighted average of the length per read calculated from RNA-seq results. Landmarks of the INTERFACE transcript are indicated. **b** The inclusion of the Elongation Switch (ES) enables accessibility-correlated differential transcriptional elongation. Two previously characterized probes, one inaccessible and one accessible[31], were tested in the presence and absence of the group I intron (black and gray, respectively), without and with the ES (left and right, respectively). **c** INTERFACE is capable of fully characterizing accessibility of an ensemble of RNA regions in a single experiment. Probes targeting gI intron regions alternating between 10 and 16 nucleotides were designed and tested. The level of accessibility per region is indicated according to the color scale below. It is interesting to note that, in contrast to chemical footprinting data typical of gI introns[77], complementary sides of stem-loops can exhibit accessibility differences (*cf*. region 9 and 13). Although discrepancies between accessibility and footprinting methods have been previously recognized[31], this observation supports a heightened sensitivity of accessibility probes to spatial arrangement of the taRNA

accessible surfaces for in vivo interactions, we designed a large-scale INTERFACE experiment to collectively characterize accessible interfaces within a native library of 71 experimentally validated sRNAs[5,6]. taRNA region sequences were selected for in vivo accessibility probing by coupling an established biophysical

model of asRNA:RNA hybridization[28] to a machine-learning algorithm (Fig. 3a). This approach simultaneously minimizes experimental effort and maximizes information recovery regarding sRNA interfaces likely to form intermolecular interactions (assumed as highly accessible regions)[28,31]. Briefly, the

**a**

Input = target RNA sequence

GCCACTGCTTTTCTTTGATGTCCCCATTTTGTGGAGCCCATC

1. Select all possible target regions

2. Predict taRNA region accessibilities
using a biophysical model.

| Target region | Accessibility prediction |
|---------------|--------------------------|
| 1 | High |
| 2 | High |
| : | : |
| 70,000 | Low |

3. Feed into machine learning algorithm.

Value of information analysis:

Bayesian prior

Probabilistic distribution

Optimization constraints
- Length
- Overlap
- Full coverage

4. "Greedily" solve the optimization problem.

Output = Suggested dynamic experimental target regions

**b**

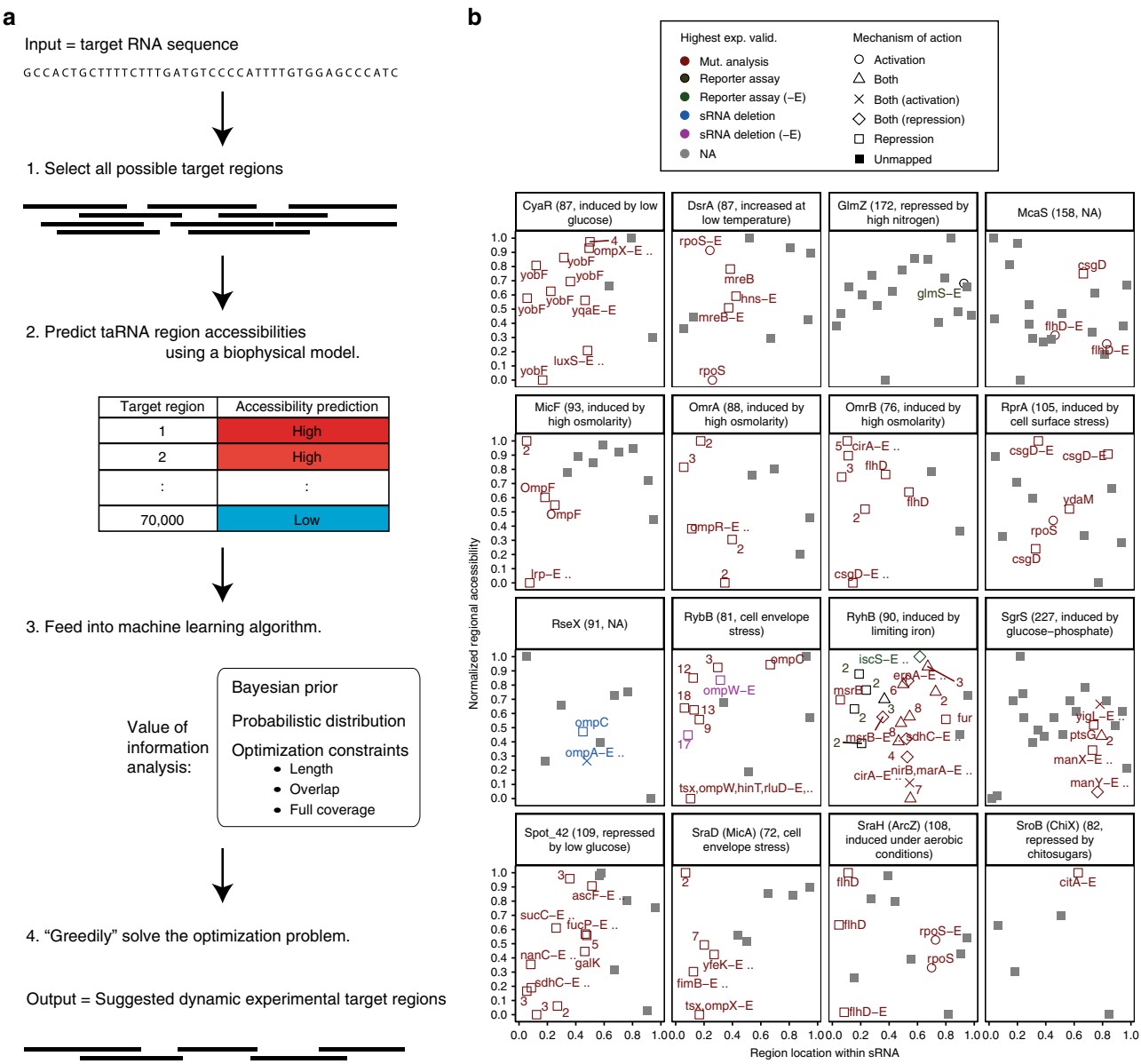

Fig. 3 INTERFACE investigation supports identification of binding site features in an ensemble of native sRNAs. **a** Workflow for machine-learning algorithm to select for accessible interfaces. Machine-learning methodologies modified for batch use were coupled to a biophysical model to select for potentially accessible regions, a consideration previously shown important to successful interactions[12, 28], in a large ensemble of *E. coli* sRNAs. Given RNA sequence only, the biophysical model predicts the accessibility of all possible regions. Regions of interest are selected via the machine learning algorithm, based on value of information analysis, to minimize experimental effort and meet specified optimization constraints. **b** The accessibility of all regions evaluated via INTERFACE, both chosen by the machine learning algorithm and deliberately designed to target previously mapped exact binding sites, are shown with respect to sRNA location for the pool of 16 well-characterized sRNAs within the ensemble (length and known stress response are indicated in the title). "-E" is listed following regions corresponding to exact mapped mRNA-binding sites that make up our exact binding site training set. The names of target mRNAs are listed beside regions that are fully contained within known binding sites (and, in the case of multiple targets, numbers corresponding to the number of mRNAs known to establish base pairs with said region). Exact binding site mRNA labels are followed by ".." if the region is also fully contained within other experimentally confirmed interactions (Supplementary Data 1). Interestingly, exact mapped binding sites tend toward the extreme thirds of accessibility and tend away from extreme 3′ positions—interactions past the first 80% of the sequence are only seen in sRNAs GlmZ, RprA, and McaS. Colors indicate the highest experimental validation (ranked as: mutational analysis, reporter assay, sRNA deletion). Shapes indicate the mechanism of sRNA regulation of mRNA expression. In the case that a region contains binding segments for multiple mRNAs, the mechanism corresponding to an exact mapped binding site is listed ("-E" in legend)

predicted accessibilities of ~70,000 randomly selected potential target regions (9–16 nt in length) along the 71 chosen sRNAs (Supplementary Data 1) were evaluated using an adapted version of a previously developed biophysical model (Methods)[28]. Predictions were fed to an algorithm known as sparse knowledge gradient (SpKG)[38,39] that is capable of delivering experimental suggestions based on value-of-information analysis[40]. Ultimately, the top 1.5% of SpKG-ranked regions (~970 sequences across all 71 RNAs), fulfilling the constraints of minimal overlap and full coverage, were chosen for INTERFACE accessibility probing to

obtain accessibility landscapes of the sRNA ensemble (Supplementary Data 2, Supplementary Fig. 3). Simulated comparisons showed that the combination of the biophysical model with SpKG reduced our experimental INTERFACE profiling efforts (Supplementary Fig. 4).

To apply INTERFACE to describe accessibility patterns of functional sRNA regions, we also deliberately selected ~40 regions previously mapped to sRNA:mRNA interactions in our training set of 16 well-characterized sRNAs for INTERFACE probing. These regions, termed "exact binding regions" are contiguous nucleotides previously confirmed to host interactions with specific mRNA targets (Supplementary Data 1). We hypothesized that these previously mapped binding sites would either be actively bound by a cellular target (appearing lowly accessible to INTERFACE asRNA probing) or unoccupied by a target and thus highly available for binding (appearing highly accessible to INTERFACE asRNA probing). Upon performing and analyzing results of the INTERFACE experiment, a comparison of proportions test confirmed that exact binding regions (38) (Fig. 3b) exhibited a significantly increased proportion in top 25% accessibility "extremities" (<0.125 or >0.875 on a 0–1 scale) compared to regions of these same sRNAs selected for INTERFACE probing by the machine learning algorithm only (179 regions) (P-value <0.1, N-1 $\chi^2$-test) (Fig. 3b). Furthermore, as expected based on prior observations[41], most of these experimentally validated sRNA sites within our training set host mRNA interactions towards their 5′-end (Fig. 3b). The exclusion of 3′-sRNA interactions may reflect the role of the Hfq chaperone Sm-protein in binding near the 3′-end (Rho-independent terminator and poly-U tail)[42], a model of regulation recently supported by ligation and sequencing approaches (RIL-seq)[25].

**INTERFACE data informs computational target mRNA prediction.** Based on trends in accessibility and position observed for known functional sRNA interfaces in the 16 well-characterized sRNAs (Fig. 3b), we anticipated that high-throughput INTERFACE data could support identification of functionally relevant regions within mechanistically uncharacterized sRNAs. As a proof of concept, we also identified regions exhibiting extreme accessibility in the 5′-end of well-characterized sRNAs that were excluded from our training set (e.g., IstR, MicL/RyeF, and RydC). Notably, many of the selected regions align with confirmed mRNA binding sites (Supplementary Data 1, Supplementary Fig. 3). We therefore hypothesized that experimentally determined regional accessibility by INTERFACE could be coupled to computational sRNA:mRNA predictions to identify mRNA targets that are most relevant to in vivo functionality. This approach is motivated by the inability of even the best-performing computational thermodynamic prediction algorithms, such as CopraRNA and IntaRNA, to account for the intracellular environment; this shortcoming often leads to low ranking of true targets (below hundreds of predicted pairings) as well as high rankings (i.e., those within the top 10 predictions) that do not correspond to any experimentally confirmed targets[12]. To aid identification of most-likely true targets, we propose a pipeline that begins with the identification of likely functional RNA regions from INTERFACE accessibility data (Fig. 4a), based on extreme (high or low) accessibility and location in the RNA. These functional regions can then be used as guides to filter results of computational sRNA target predictions, to obtain a reduced list of computational targets (Fig. 4a, left of table). Importantly, the filtered list includes targets of any computed energy rank, as long as their predicted sRNA binding region was found to exhibit extreme accessibility in vivo by INTERFACE.

To benchmark this approach, we first considered three representative uncharacterized sRNAs (SroE[43], SroG[43], and Tpke70[44]) for interaction validation by in vitro electrophoretic mobility shift assays (EMSAs) (Supplementary Data 3). These sRNAs exemplify a variety of length, hybridization profiles, and reported Hfq dependencies (Supplementary Data 1). Taking extreme accessibility and position (5′ and 3′ ends) as a signature of functional sRNA interfaces, up to three representative likely functional regions were identified for each sRNA (Methods) (Fig. 4b). Notably, we considered functional regions in the 3′ position to include sRNAs whose characterization has likely been limited by an inability to co-immunoprecipitate with Hfq (ie. Tpke70[45]). In cases in which an sRNA region's likelihood of being a functional interface was ambiguous based on accessibilities, we paired the accessibility signature with the YUNR motif previously identified as characteristic of asRNA:RNA recognition[46] (Methods). INTERFACE-informed mRNA targets were then selected as those within the top 100 IntaRNA mRNA target predictions[11] whose predicted interaction relied on at least five nucleotides of the likely functional sRNA region (Methods). IntaRNA[11] was selected for computational predictions as it offers high accuracy without requiring homology information[12]. We benchmarked INTERFACE-informed predictions against conventional methods of putative target selection for validation (e.g., selection of the top-ranked computational targets) (Fig. 4a) (Supplementary Data 3). All mRNA targets considered for experimental validation were additionally subject to in vivo expression constraints (Methods) in which we excluded mRNA candidates whose predicted sRNA binding site was not expressed in previously performed transcriptomics studies[47].

Upon biochemical experimentation (EMSA) of predicted targets, we found that positive predictive values (PPVs) of INTERFACE-informed and top-ranked computational predictions for each sRNA are comparable, but positive mRNA targets added by INTERFACE-informed predictions are often low-ranking in IntaRNA predictions, and thus have low chances of being further considered as true targets in the absence of other experimental data (Fig. 4c, rank indicated by dotted lines). The ability of the INTERFACE-informed approach to capture computationally lowly ranked true mRNA targets is best exemplified in the case of SroG sRNA (Fig. 4c, Table 1). Specifically, in addition to the top-ranked targets suggested by IntaRNA, INTERFACE-informed selections also capture true mRNA targets ranking as low as 64/100, 75/100, and 76/100 (Fig. 4C, Table 1). Positive binding EMSAs and predicted interactions of all validated sRNA:mRNA pairs for these sRNAs are illustrated in Fig. 5. We also tested the use of INTERFACE-informed predictions to identify targets for three other under-characterized sRNAs, SroA, SroH, and Tpke11, confirming *rhtA* and *rplK* for SroA, *kbl* for SroH and three targets (*pgsA*, *prlC*, *yehU*) for Tpke11 (Supplementary Data 3 and Supplementary Fig. 5). Finally, we evaluated whether INTERFACE could identify new targets in well-characterized sRNAs by suggesting regions of extreme accessibility not already implicated in known target binding. As an example, we observed extreme accessibility in CyaR and GcvB (CyaR-13 and GcvB-21, respectively, Supplementary Data 2, Supplementary Fig. 3), which are outside of previously characterized binding regions (Supplementary Data 1). Importantly, using these regions to inform computational predictions (Methods), we confirmed additional targets *zapC* for CyaR and *thrL* (previously validated as an in vivo target of the *Salmonella* GcvB homologue[48]) and *gadX* for GcvB (Supplementary Fig. 5, Supplementary Data 3). Interestingly, both *gadX* and GcvB are known contributors to acid resistance but their function has not previously been coupled in vivo[49].

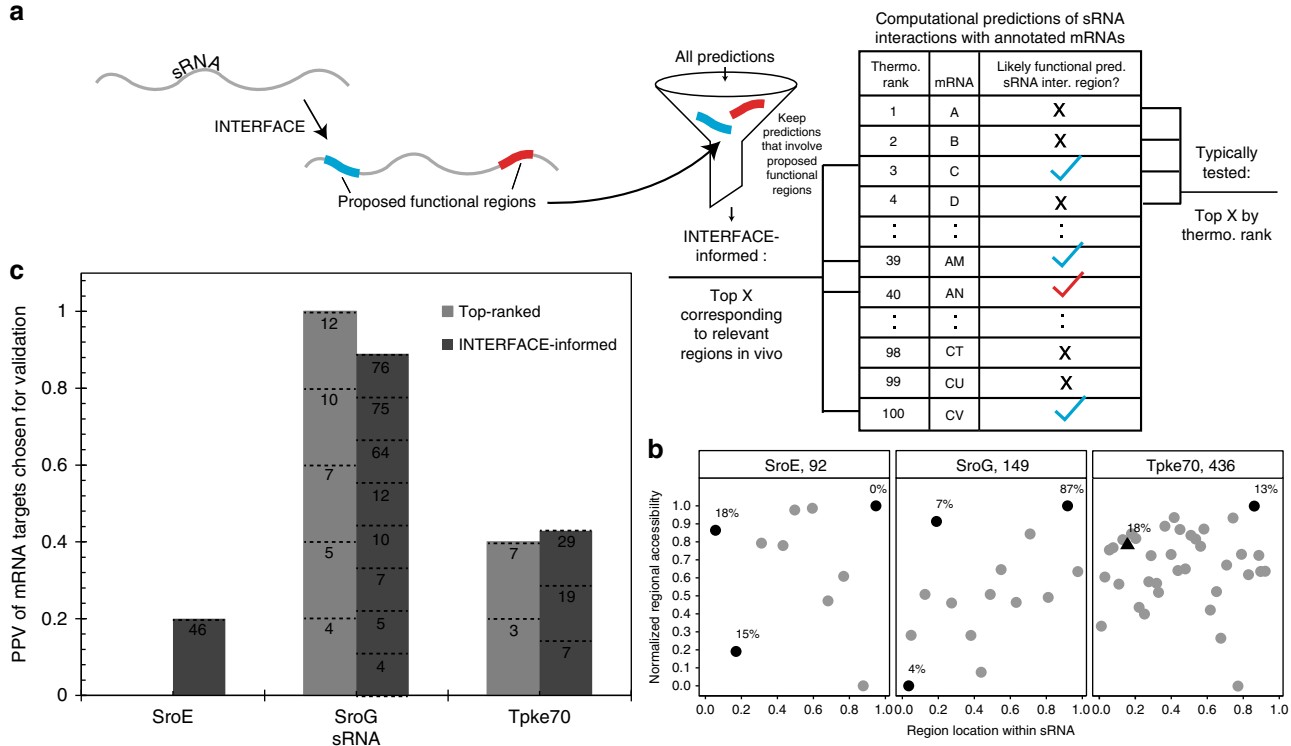

**Fig. 4** Benchmarking the use of INTERFACE for selection of "true" predicted target mRNAs of under-characterized sRNAs. **a** INTERFACE data, along with position and motif information, allows filtration of a pool of sRNA regions targeted for the selection of likely functional regions. Specifically, likely functional regions are chosen as those that exhibit accessibility values within extremes chosen for benchmarking (>0.75 or <0.25), 5'- or 3'-end position, and, in rare circumstances, the presence of a YUNR motif (see Methods). These likely functional regions can be used to inform computational mRNA target predictions for sRNAs to suggest positive binding partners (INTERFACE-informed, left of table) that may not typically be chosen for experimental validation (top-ranked, right of table) due to low energetic favorability. **b** Regional position versus accessibility for three representative uncharacterized sRNAs selected for benchmarking of computational mRNA predictions. Dark gray regions correspond to likely functional regions, chosen for their extreme accessibilities in the 5' or 3' 20% end of the transcript. In the case of accessibility clustering, presence of a YUNR motif is indicated by a triangle. Percentages of predicted mRNA target interactions within top 100 computational predictions that involve the likely functional regions are listed beside the corresponding region. **c** Positive predictive values (PPV) of computationally predicted binding partners for three representative uncharacterized sRNAs as evaluated using EMSA. Numbers within bars correspond to the ranking (out of 100) of all confirmed mRNA interaction partners for each prediction category (INTERFACE-informed v. top-ranked). For SroE, one mRNA target from the INTERFACE-informed group (ranking 46/100 in IntaRNA predictions) was confirmed via EMSA, although none of the top-ranked IntaRNA predicted targets tested (within top 12/100) exhibit sRNA binding (Supplementary Fig. 5). For SroG, INTERFACE correctly pinpoints a region implicated in many of the top predictions (Fig. 3b), further giving confidence to these as true partners, and also pinpoints lowly ranked targets via the two proposed functional regions at the 5'-end (Fig. 3b). Finally, considerable overlap between INTERFACE-informed predictions and top-ranked tested IntaRNA predictions (within top 14/100 rank) of Tpke70 led to the confirmation of *mcrA* as a target (Table 1). Importantly, IntaRNA correctly predicts a confirmed weak target (*agaS*) to interact with a region not captured by the INTERFACE-informed group

**INTERFACE uncovers regulation at the GlmY terminator.** Upon analysis of sRNA profiles, we noticed a unique pattern in GlmY, in which accessibility drastically increases in the terminator region (Fig. 6a). In light of the possibility of these regions being regulatory, we collected top 100 IntaRNA predictions corresponding to the full sRNA sequence based on overlap with the accessible terminator regions. We additionally evaluated predictions obtained by inputting the accessible portion of the terminator region alone as the query RNA, given our suspicion that predicted interactions involving the terminator would not be well-represented given strong terminator structures. Of seven total in vitro-tested targets, we observed strong binding of GlmY to two zinc-binding mRNAs, *add* and *yphC*[50] (Fig. 6b); of these, one was computationally predicted using the entire GlmY sequence as the query input. Importantly, we confirmed terminator-dependent binding of these confirmed targets (Fig. 6c) via EMSAs in the presence and absence of the terminator region (Fig. 6c, Supplementary Data 3). Given that GlmY has only been implicated in sequestration of the RapZ protein[33], this is the first report of GlmY engaging in direct RNA:RNA interaction.

Notably, regulatory sRNA:mRNA base-pairing involving sRNA terminators has been observed[51,52], but remains an uncommon mechanism in the literature. It is worth noting that we did not find any dependence on the presence of the terminator region for target binding in the cases of CyaR and GcvB, despite the fact that these newly uncovered targets were also predicted to bind terminator regions (Supplementary Fig. 5).

**Hfq selectively impacts sRNA accessibility landscapes.** The INTERFACE accessibility of all regions within the 71 sRNAs was additionally evaluated in an isogenic BW25113 Hfq-knockout strain (JW4130-1[32]) to evaluate the global impact of Hfq on regional hybridization within a library of sRNAs in their native intracellular context (Supplementary Data 2). These results were analyzed relative to the accessibility profiles of these sRNAs in the wild type strain (in the presence of Hfq). Based on earlier literature reports, the 71 sRNAs represent 39 Hfq-dependent RNAs (with less than half mechanistically well-understood), and 13 Hfq-independent RNAs, with the remainder of the sRNAs

**Table 1 Rank, name, and in vitro binding results for three representative uncharacterized sRNAs**

| Rank | mRNA | INTERFACE-informed or top-ranked? | In vitro binding? |
|---|---|---|---|
| SroE | | | |
| 3 | yieH | Top-ranked | No |
| 5 | yeaG | Top-ranked | No |
| 9 | yahN | Top-ranked | No |
| 10 | mdtA | Top-ranked | No |
| 12 | creA | Top-ranked | No |
| 18 | arcC | INTERFACE | No |
| 32 | pdxY | INTERFACE | No |
| 38 | marC | INTERFACE | No |
| 46 | ligB | INTERFACE | Yes (weak) |
| 61 | fadH | INTERFACE | No |
| SroG | | | |
| 4 | cdsA | Both | Yes |
| 5 | cysJ | Both | Yes |
| 7 | yfbR | Both | Yes |
| 10 | yncD | Both | Yes |
| 12 | mepM | Both | Yes |
| 64 | potA | INTERFACE | Yes |
| 65 | dosP | INTERFACE | No |
| 75 | pgl | INTERFACE | Yes |
| 76 | fecE | INTERFACE | Yes (weak) |
| Tpke70 | | | |
| 3 | agaS | Top-ranked | Yes (weak) |
| 6 | dkgB | Both | No |
| 7 | mcrA | Both | Yes |
| 12 | mdoB | Both | No |
| 13 | dbpA | Both | No |
| 14 | degS | Both | No |
| 19 | torT | INTERFACE | Yes |
| 29 | glnK | INTERFACE | Yes |

uncharacterized in terms of functional Hfq dependency (Supplementary Data 4). When comparing INTERFACE datasets derived from the *hfq*-null and *hfq*+ strains, we anticipated strongly altered sRNA regional accessibilities for sRNAs that highly depend on Hfq for target regulation. Consistent with previous reports, perturbations of accessibility (*P*-value <0.1, paired two-tailed *t*-test) were observed in distinct regions of the well-established Hfq-dependent sRNA Spot 42 (Fig. 7a) but were not seen in RybA (MntS) sRNA, which was shown not to co-immunoprecipitate with Hfq in a prior study (Fig. 7b)[53].

In light of the observed sensitivity of INTERFACE to Hfq-associated accessibility landscape changes, we propose a binary INTERFACE-derived Hfq dependency for all sRNAs under investigation, classifying 46 as Hfq-dependent due to significant variation in their ability to hybridize with cognate asRNA probes, *P*-value <0.05 for paired two-tailed *t*-test, in at least one region in the absence of Hfq. Within this group (106 regions), there was a clear positive skew of hybridization changes in *the hfq*+ strain relative to the *hfq*-null mutant (~2-fold) (Fig. 7c), supporting the previously proposed structure-relaxing Hfq RNA chaperone mechanism[54–56]. It is likely that INTERFACE is sensitive to structural rearrangements that can be attributed to the presence of Hfq in vivo. Furthermore, as seen within native RNA:RNA interactions, some INTERFACE asRNA:target interactions may also be facilitated by Hfq. For example, absent Hfq, it is interesting to note a significant decrease of accessibility in GlmY terminator (GlmY-17, Supplementary Data 2), suggesting Hfq may facilitate terminator-involving GlmY:mRNA interactions in vivo, both with the INTERFACE probe and the actual mRNA target. Although many sRNAs probed in this work exhibit Hfq-

dependent hybridization patterns, many others show no hybridization landscape changes due to Hfq absence, suggesting that their RNA-binding activity is independent from Hfq under tested conditions. Supplementary Data 4 combines our INTERACE-inferred Hfq dependencies with corresponding experimental confirmations reported in the literature and also includes other information related to Hfq dependency, e.g., sRNA class (I or II)[57], ProQ protein associations[16], etc. Notably, INTERFACE-based classification is consistent with 31 of 52 sRNAs whose Hfq association (or lack thereof) has been previously experimentally characterized (Supplementary Data 4).

## Discussion

In this work, we detail the design and validation of INTERFACE, a synthetic transcription elongation-based reporter system that supports simultaneous in vivo sensing of accessible interfaces within an ensemble of user-defined RNA regions via RNA-seq. INTERFACE enables determination of the "accessosome" which, in contrast to the "interactome"[23–25] and "structurome,"[26] is a measure of the functionality, i.e., the capacity to interact, of a large number of regions in the transcriptome (Fig. 8).

Overall, we demonstrate that INTERFACE can support further mechanistic understanding of regulatory RNA networks by: (i) capturing hybridization-based dependencies on intracellular regulatory factors, (ii) pinpointing sites capable of creating RNA:RNA interactions, and (iii) identifying regions that can be involved in mRNA targeting for uncharacterized (and even well-characterized RNAs) to propose novel RNA regulatory potential. Collectively, these findings demonstrate the advantage of obtaining large-scale in vivo accessibility data to complement computational prediction algorithms that aim at uncovering in vivo RNA:RNA interactions. This approach is a viable high-throughput alternative to effective but lower throughput experimentation that is typically necessary to inform computational predictions (e.g., co-immunoprecipitation approaches)[58] for groups of sRNAs that cannot be analyzed via recently developed high-throughput techniques (RIL-seq[25], CLASH[24], or modified CLASH[23]) due to low expression or functional independence from chaperone or degradation proteins.

Interestingly, the target repertoire of some of the studied sRNAs suggests that they may have important roles in stress responses such as metal ion binding and transport (SroG targets *yncD*, *mepM*, and *fecE*), and DNA ligation (SroE target *ligB*)[50]. These results suggest that INTERFACE may be valuable in revealing new stress-responsive sRNA networks. The ability of INTERFACE in revealing novel regulation is further exemplified in GlmY, whose high-terminator accessibility prompted the in vitro validation of two novel mRNA binding partners (RNAs coding for Zn-binding proteins:[50] *add* and *yphC*) that depend on the terminator sequence of GlmY for interactions. To date, sRNA terminator activity has been seen only in rare instances; for example, *chb* mRNA antagonizes the ChiX:*chiP* sRNA:mRNA interaction by pairing with the ChiX terminator, decreasing ChiX levels and activity[52]. Furthermore, rare RprA regulation of *csgD* relies on three distinct regions, one of which is in the terminator; however, mutation to the terminator region alone does not affect in vivo target repression[51]. It is not clear whether these antisense GlmY-TER:mRNA interactions detected by INTERFACE represent an mRNA feedback, or whether the terminators, like other RNA stem-loops, are capable of initiating antisense pairing and regulation.

INTERFACE may also offer insights into sRNA:mRNA binding initiation, as supported by its sensitivity to very subtle shifts in the target region window. Specifically, regions contained within previously mapped mRNA-binding sites sometimes have vastly

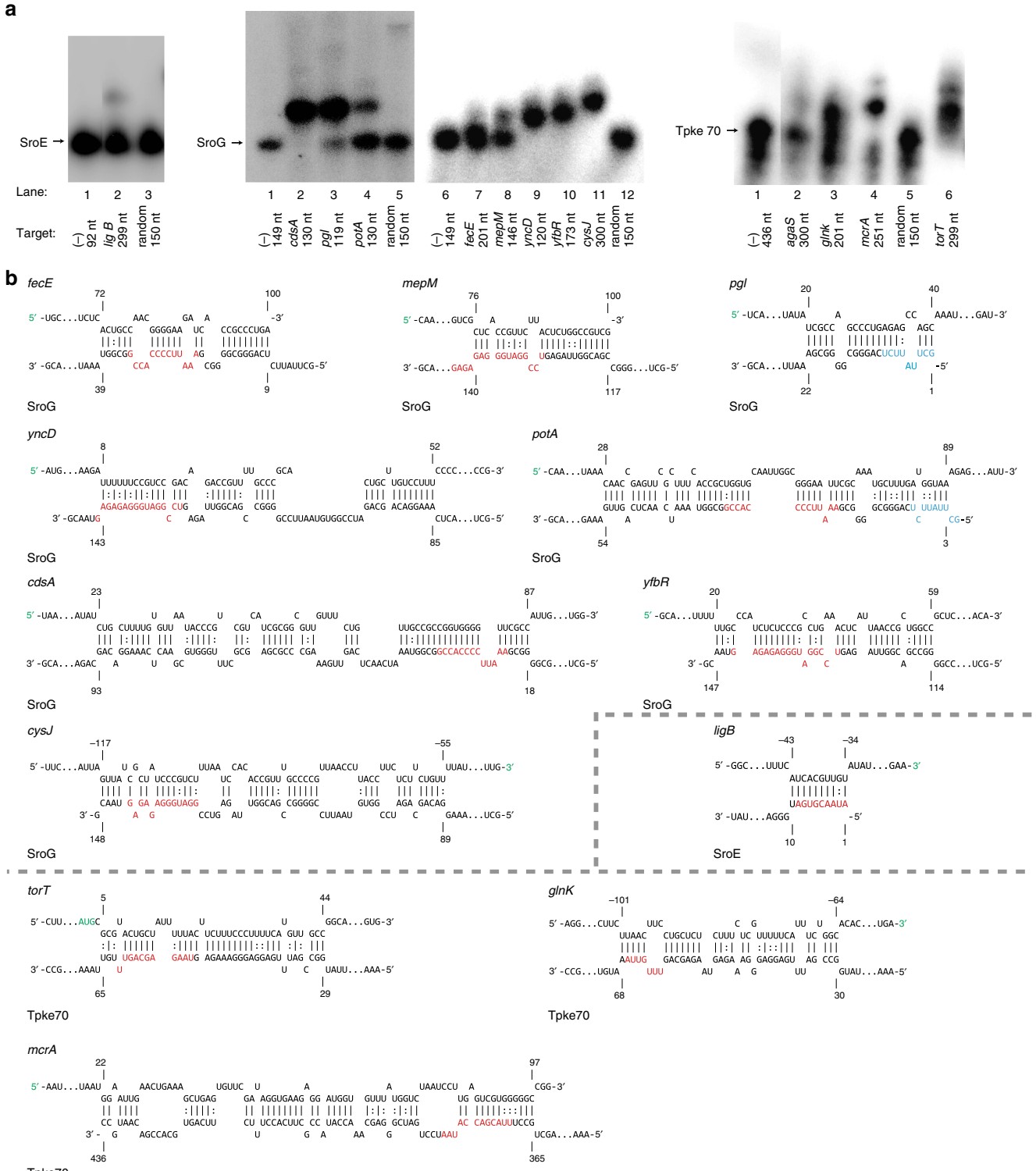

**Fig. 5** Validated sRNA:mRNA interactions support the use of INTERFACE insights to inform sRNA target predictions. **a** All positive EMSA results from the benchmarking study are pictured with respective controls (i.e., sRNA only and sRNA+ random sequence). A 1:10 molar sRNA:mRNA ratio was used in all cases. sRNA names are indicated above corresponding EMSA results, and mRNA names and lengths of representative fragments chosen for validation are indicated under each respective lane. **b** Predicted IntaRNA interactions are shown for each positive result. Likely functional regions that were selected via the pipeline for selection of "true" predicted mRNA targets of under-characterized sRNAs (Fig. 4b) are highlighted in colors corresponding to their observed accessibilities (red = high, blue = low). mRNA positions are indication, in which the 5'-UTR corresponds to negative position values relative to the start codon. Start codon positions (first nucleotide = +1) are highlighted in green

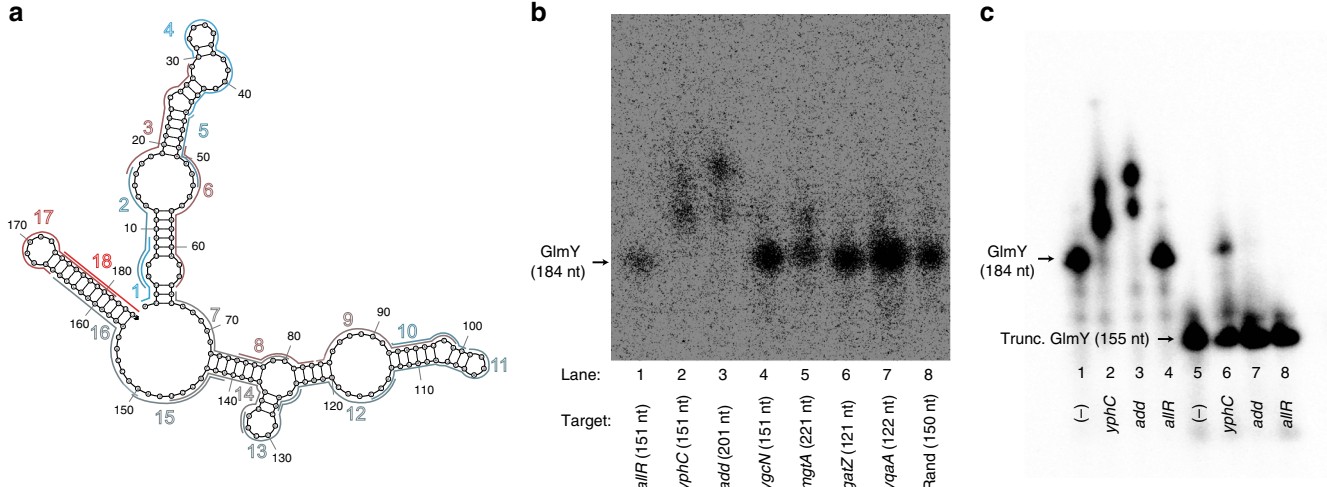

**Fig. 6** INTERFACE pinpoints mRNA regulation in the GlmY terminator region. **a** Regions 17 and 18 within the GlmY terminator exhibit drastically high accessibility compared to all other regions probed for accessibility within the sRNA. Pictured minimum free energy secondary structure was predicted via Nupack[78]. **b** Seven mRNAs were tested for interaction with GlmY based on computational predictions implicating at least five nucleotides of the accessible terminator regions in target binding. Importantly, *allR* and *yphC* were selected from IntaRNA predictions of the INTERFACE–accessible terminator sequence alone as the query RNA input. Two strong targets are observed: *yphC* and *add*. **c** Binding assays of the two confirmed strong targets as well as a negative control (*allR*) were performed with full-length GlmY as well as a truncated version, in which all nucleotides downstream of the first basepair in the terminator stem were excised. Confirmed target binding to GlmY is diminished in the absence of the terminator

different accessibilities than regions with slightly different indices; for example, the DsrA:*rpoS* mapped binding site (23 nt) has relatively high-normalized accessibility (>0.9, region 3) compared to a 16 nt stretch completely contained within this mapped site (zero accessibility, region 4) (Fig. 3b, Supplementary Fig. 3). As the contained region lacks five nucleotides within the 5′ loop of the mapped site, it appears that INTERFACE may be sensitive to structural features of binding sites such as native toeholds, or seed regions that support RNA unfolding when targeted[11,36]. Another example of this phenomenon is seen with the *csgD* binding site within RprA (Fig. 3b, *cf*. regions 5 and 6) (Supplementary Fig. 3). These observations speak to the importance of probing the full length of mapped binding sites, or, if such sites are unknown, increasing the sample space by interrogating overlapping or partially redundant stretches with representative lengths (e.g., the average length of known sRNA:mRNA interaction sites, or guided by free energy considerations).

Further, approximately two-thirds of all sRNAs analyzed herein by INTERFACE show a pattern of Hfq-dependence for hybridization, including a handful reported as Hfq-independent or previously uncharacterized (18 sRNAs) (Supplementary Data 4). Within this subset of sRNAs, our results support prior mechanistic observations of Hfq acting as a chaperone to selectively rearrange the RNA to increase favorability of some sRNA: mRNA interactions (Fig. 7c), likely via unfolding[15]. Interestingly, no consistent magnitude or direction of accessibility shifts between the *hfq*-null and *hfq*+ strains was observed within distinct sRNA regions known to establish direct interaction with Hfq sites (Supplementary Data 1). These data are further consistent with the aforementioned model in which Hfq binds sRNAs at a few characteristic short motifs without occluding the base-pairing surface[42]. Furthermore, and contrary to general acceptance[3], many *cis*-encoded sRNAs (ten) also have accessibility landscapes that are affected by Hfq deletion (Supp. Table 4), suggesting either that they are affected indirectly or that Hfq has a less discriminatory scope of regulation, as has been suggested of the ProQ chaperone[59]. In contrast, several sRNAs were shown to be completely functionally independent of Hfq (23 sRNAs, Supplementary Data 4). For some of these sRNAs, this independence

may be attributed to the environmental conditions under which probing was performed, as Hfq availability is limited[13,14]. For others, it is possible that they are regulated via other chaperone proteins besides Hfq, such as FinO domain-containing or cold shock proteins[14]. To this end, investigating protein associations of sRNAs that appear unaffected by Hfq may prove informative.

We anticipate the utility of INTERFACE in characterizing unique hybridization behaviors of other distinct classes of RNAs (tRNAs, mRNAs, etc.) as well as investigating hybridization changes in response to various environmental conditions to support identification of stress responsive regulatory RNAs and associated functional regions. More broadly, given that INTERFACE mostly exploits formation and disruption of Watson-Crick base pairing, we foresee that INTERFACE will support investigation of RNA function in other organisms in vivo, pending appropriate replacement of the anti-termination (AT) mechanism. Importantly, TnaC peptides have been identified in species of many pathogenically relevant bacterial genera (e.g., *Vibrio*, *Yersinia*, *Shigella*)[60], suggesting that they share similar ribosomal stalling-based AT mechanisms with *E. coli*. Furthermore, as eukaryotic RNA polymerases have been shown sensitive to hairpin terminators in vitro[61,62], an adaptation in which the stability of the hairpin structure is determined by the extent of asRNA binding to its target region could be viable for use in higher organisms. Combining high-throughput information about regulatory RNA regions, intracellular factor dependency, and environment-associated activity within a multitude of RNAs may be key to the design of synthetic antisense RNA or other regulatory molecules.

## Methods

**Plasmids and strains**. Three different *E. coli* strains were used in this work: K-12 MG1655 for the experiments performed via overexpression of the gI intron to establish the technique, BW25113 (Keio collection parent strain[32]) and the Hfq-deficient isogenic strain (JW4130-1Δ*hfq* from the Keio collection[32]) for experiments performed to characterize accessible interfaces in native RNAs. The Hfq-deficient strain was cured of a kanamycin resistance gene cassette following a FLP recombination protocol[32] and the resistance deletion was confirmed via genomic PCR. Two main plasmids were constructed from iRS³GG and modified iRS³GG (modified for targeting native RNAs) plasmids, published previously[28], for this

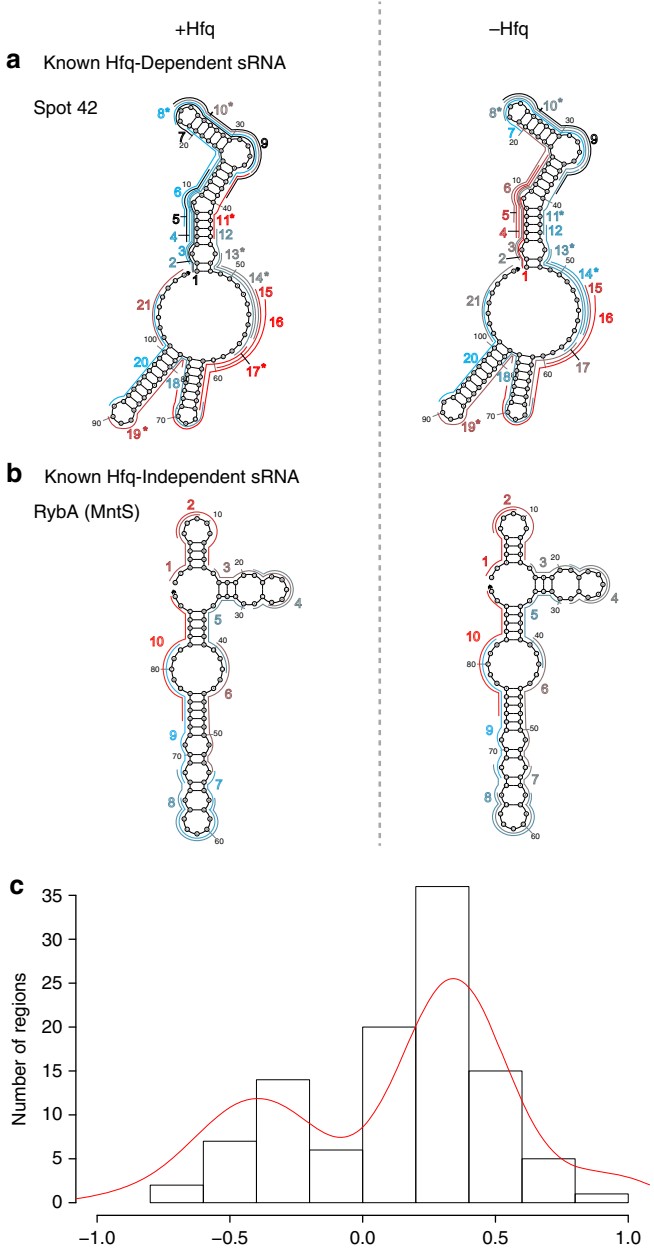

**Fig. 7** INTERFACE can distinguish sRNA accessibility changes attributed to presence of a pleiotrophic RNA regulator. **a** INTERFACE is sensitive to Hfq dependency, as shown by changes in accessibility landscapes in a strongly Hfq-dependent sRNA (Supplementary Data 4) Spot 42 between the parent strain (+Hfq) and an isogenic Δ*hfq* strain (−Hfq). Accessibility differences significant to *P*-value <0.1 in a paired two-tailed *t*-test are indicated by asterisks. **b** In contrast, no significant changes are observed for RybA, an sRNA believed to be Hfq-independent due to its inability to co-immunoprecipitate with Hfq in prior studies (Supplementary Data 4). The level of accessibility per region is indicated according to color (blue = low, gray = mid, red = high). **c** INTERFACE supports an accessibility-increasing role of Hfq in sRNAs, indicated by the skew of accessibility difference upon presence of Hfq relative to the absence of Hfq (parent − Δ*hfq*) to the right for all significant differences (*P*-value <0.05, two-tailed *t*-test) within the entire sRNA ensemble (106 regions)

work: Overexpression-INTERFACE (O-INTERFACE, for overexpressed taRNAs) and INTERFACE (for native taRNAs), respectively (plasmid maps in Supplementary Fig. 1A and 1B; annotated plasmid sequences in Supplementary Data 5 and 6, respectively). These plasmid constructs mainly differ from their corresponding parent plasmids in containing (1) an adaptor, containing *tnaC* and a *rut* site, and (2) a truncated version of a (GFP) reporter gene in which start and stop codons have been preserved to allow for transcript size characterizations.

**Estimation of binding potential using a biophysical model**. In order to efficiently design asRNA probes for 71 sRNAs, we employed, in combination with a machine learning algorithm (described below), an un-optimized version of a model previously reported[28] to explain hybridization efficacy, *v*, as follows:

$$v = \bar{\theta}(\Delta G_{\mathrm{tf}} - \Delta G_{\mathrm{asT}}) + \Delta G_{\mathrm{asF}}. \tag{1}$$

In this model, the Δ*G* terms represent the free energies which must be considered for the interaction of the target region with the reporter probe, in which subscripts "tf," "asT," and "asF" represent target unfolding, binding between the asRNA and target, and the asRNA reporter probe unfolding, respectively. This model also includes a pseudo-frequency factor ($\bar{\theta}$) to account for the global ensemble of structures within the target region. This term is evaluated at a regional level and is thus calculated as the summation of each nucleotide's local unpaired probability over the length of the target region, as estimated by base-pairing probabilities, using the "AllSub" subroutine in the RNA-structure webserver[63–65].

**Machine learning algorithm**. To optimally select for accessible regions in 71 experimentally confirmed bacterial small RNAs in *E. coli*, we adapted a machine-learning algorithm called sparse knowledge gradient (SpKG)[38,39] to a weighted set cover problem. The SpKG algorithm is developed to solve the sequential ranking and selection problem, in which, at each iteration, one or several experimental suggestions are provided based on value-of-information analysis by taking into account the new observations (biophysical model predictions). SpKG can be used to adaptively select the targeted regions within a large molecule to identify which regions are more amenable to establish interactions with other molecules[66]. By adapting the SpKG algorithm to a weighted set cover problem, we attempt to maximize the value of information of all suggested probes under the constraints of (i) full taRNA regional coverage of each sRNA molecule and (ii) minimum overlap between suggested probes. In this way, this work is the first demonstration of applying SpKG in a batch setting (as opposed to a sequential setting).

In the following, we provide the mathematical formulation of the problem. For any RNA molecule with length *L*, suppose there are *n* potential taRNA regions with the length specification (9–16 nucleotides). Let $[i_1,j_1],[i_2,j_2],...,[i_{1n}j_n]$ be intervals on $[1,L]$ that denote these *n* potential target regions. Let $x_1,x_2,...,x_n \in \{0.1\}$ be binary variables that denote either the *k*th target region is selected or not. We use $v_k$, $k = 1,...,n$ to represent the knowledge gradient value of the *k*th target region. These values can be computed via the SpKG algorithm[38]. Our optimization problem can be written as

$$\max\left(\sum_{k=1}^{n} v_k x_k - \lambda \sum_{k=1}^{n} x_k\right). \tag{2}$$

$$\mathrm{s.t.} \bigcup_{\{k:x_k=1\}} [i_k, j_k] = [1, L].$$
$$x_k \in \{0, 1\} \text{ for all } k = 1, \dots, n.$$

Here *λ* is a tunable parameter that penalizes the number of target regions selected to minimize overlapping. As described, this optimization problem is a weighted set cover problem, contained within Karp's 21 NP–complete problems[67], and cannot be solved in polynomial time. In order to solve it efficiently, we use a Greedy algorithm[68] to approximate a solution.

Algorithm:

1. $C \leftarrow \emptyset, I \leftarrow \emptyset$. (Here *C* is the set of nucleotides covered so far; *I* is the set of index for the selected targeting regions.)
2. While $C \neq [1, L]$ do

 a. for all $k \notin I$, let $\alpha_k = \frac{\lambda - v_k}{|[i_k j_k] - C|}$
 b. choose $k^* = \mathrm{argmin}\, \alpha_k$
 c. update $C \leftarrow C \cup [i_k j_k], I \leftarrow I \cup \{k^*\}$.

3. Output.

**Computational simulations to test for algorithm performance**. To quantify the performance of this algorithm in synthetic simulations, we compare it with two comparatively naive algorithms for designing aRNA probes—exploration and exploitation. The exploration (uniform) algorithm selects random taRNA regions with a uniform length of 12 nt for each RNA molecule. The exploitation algorithm

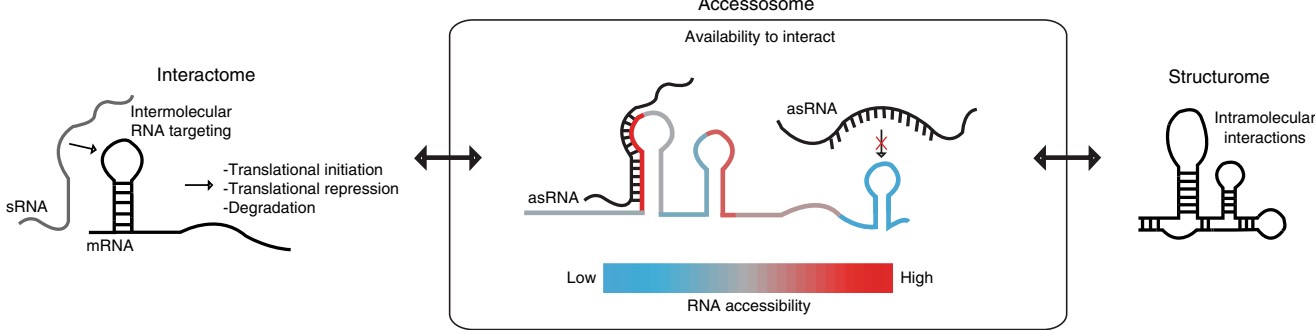

**Fig. 8** INTERFACE sits at the core of the structure–function relationship. By sensing functional hybridization differences attributed to both in vivo interactions (interactome, left) and secondary structure (structurome, right), the accessosome offered by INTERFACE sits at the core of the structure–function relationship

selects target regions using the aforementioned biophysical model with identical coverage and overlap constraints as the SpKG algorithm, but without any machine learning optimization. In these controlled synthetic simulations, we sample the "true" accessibility coefficients from a stochastic process. Taking into account simulated noise of each experiment, we then normally sample the observations from the three sets of target regions generated using the SpKG, exploration (uniform), and exploitation algorithms. For each set of suggested target regions, we consider a metric called experimental effort to estimate "how close" we are in terms of identifying the most accessible target region in a given number of simulation iterations (Supplementary Fig. 4). In this case, the metric considers, for each sRNA, the difference between the observed maximum accessibility and the "true" accessibility of the optimal region (expected to exhibit the highest accessibility in the molecule) chosen by each respective algorithm. The experimental effort for each algorithm is rescaled from zero to one.

**Synthesis of constructs**. All probes were inserted into one of two INTERFACE plasmids for hybridization evaluation of regions within an overexpressed gI intron taRNA (O-INTERFACE) or a native taRNA (INTERFACE), via golden gate (GG) cloning[28]. Specifically, 30 designed probes targeting the gI intron were cloned into the O-INTERFACE plasmid. Additionally, five previously designed[31] probes (original-4, 6, 7, 9, and 10) were introduced into both the O-INTERFACE plasmid and a corresponding modified plasmid lacking the elongation switch completely (Fig. 3b). For the synthesis of the >900-probe library (sRtar) targeting 71 sRNAs in *E. coli*, a high-throughput version of GG cloning was employed, in which up to 10 sets of separately annealed primers were combined for cloning into the INTERFACE plasmid. Cloning for INTERFACE plasmid libraries targeting regions within DsrA or RyhB was done separately, similarly in high-throughput fashion. A "no interaction" probe (a randomized 15-mer tested for minimum complementarity to genome in *E. coli*) was incorporated into both INTERFACE constructs to generate the sequestered RBS control[69]. For the free RBS control only, a randomized RSE was introduced to the sequestered RBS control via Gibson assembly to prevent formation of the stem-loop that serves to block the RBS[69].

All INTERFACE constructs, including ~100 sRtar DNA library pools, were transformed initially into *E. coli* electro-competent cells (Turbo, NEB) and, upon plasmid extraction, into the appropriate experimental strain. Each LB-agar plate containing the sRtar library (~100 plates) was supplemented with small LB volumes and thoroughly scraped using cell spreaders. The resulting combined cell mass was recovered for 1.5 h at 37 °C and stored at −80 °C upon supplementation with 30% (v/v) glycerol for a final 15% (v/v) glycerolized culture. INTERFACE strain libraries containing DsrA and RyhB-specific asRNA probes were stored separately.

For strain confirmation, each O-INTERFACE or control INTERFACE plasmid was individually sequenced upon plasmid extraction from the relevant experimental strain (K-12 MG1655). For the sRtar library, strain diversity was confirmed by sequencing randomly selected subsets of colonies per plate (>60 total). sRtar library diversity was further supported by RNA-seq analysis, allowing confirmation of >95 and >98% of clones for parent and Δhfq strains, respectively. The inability to confirm a small percentage of clones was likely due to limited sequencing depth. All synthesized constructs with their corresponding probe sequence, taRNA, and primers used are listed in Supplementary Data 7.

**INTERFACE experiments**. INTERFACE experiments were performed following a previously reported protocol for a related low-throughput regional hybridization quantifying system, in which GFP fluorescence is the system output[31,69]. For the experiments using the O-INTERFACE plasmid to validate the system, we prepared individual overnight cultures (biologically independent samples) for each construct. Equal parts of each resulting culture were combined and seeded into 40 mL of LB to comprise 1% of total volume. Cell cultures were grown in triplicate under four different induction conditions (N = 12): (1) no anhydrotetracycline (aTc), no

arabinose (ara); (2) 20 μL of aTc (final concentration: 100 ng/μL), no ara; (3) no aTc, 800 μL of 20% ara (final concentration: 0.8%) and (4) 20 μL aTc, 800 μL 20% ara.

For the experiment intended to characterize native taRNAs (INTERFACE plasmid), 100 mL of LB were seeded with 600 μL of the sRtar library directly from gradually thawed freezer stocks in triplicates. Control probes were seeded into these triplicates cultures separately as follows (1) 400 μL free RBS (from an overnight culture), (2) 400 μL sequestered RBS (from an overnight culture), and (3) 400 μL each of the libraries of probes targeting DsrA and RyhB (from individually stored freezer stocks). Kanamycin was added to all cultures to obtain a final concentration of 50 μg/mL. Samples were grown in triplicate under two induction conditions (no ara and 2 mL of 20% ara), and in two separate strains (parent and Δhfq) for a total of twelve samples. Samples were induced 1–1.5 h post seeding (OD ~0.2–0.3), recovered 5 h post induction (OD ~0.5–0.7), and immediately processed for total RNA extraction.

**Total RNA extraction**. RNA extractions were performed according to established techniques (see Supplementary Materials and Methods).

**Synthesis of DNA libraries for next generation sequencing**. Following RNA extraction, and in preparation for RNA-seq, we used the NEBNext Multiplex Small RNA Library Prep set for Illumina (NEB E7330) to prepare the DNA libraries. The RNA Fragmentation step was omitted to guarantee that each 5′-read and 3′-read from RNA-seq could be reliably assumed as the true starts and ends, respectively, of the corresponding transcripts. The protocol provided for preparation of DNA libraries by the supplier (NEB) was followed with a few adaptions. Briefly, a 1:2 dilution of the 3′ SR adaptor for Illumina was ligated overnight (18 h at 16 °C) using between 0.5 μg and 1 μg of non-fragmented RNA as the starting material. Next, the SR RT Primer for Illumina was annealed to the 3′ adaptor ligated RNA samples and then the 5′ SR adaptor for Illumina was ligated (1 h at 25 °C). Subsequently, a reverse transcription reaction was performed (60 min at 50 °C) to obtain cDNA, which was immediately enriched via a standard PCR amplification as recommended by the supplier with a modified extension time of 1 min per cycle instead of 15 s for a total of 15 cycles. The resulting DNA was purified using the AMPure Bead XP system and a magnetic rack in at least two wash cycles with freshly prepared 80% ethanol.

**Illumina sequencing of DNA libraries**. DNA libraries were submitted to the GSAF core facility (UT Austin) for sequencing. The samples were analyzed for their size distribution using a bioanalyzer (Agilent). To enrich for the transcripts of interest (INTERFACE plasmid transcripts), in the case of the sRNA experiment, and enhance mapping depth of every single probe, the GSAF facilities performed a Pippin Prep (Sage Science) preferentially selecting for transcript sizes between 120 (exact length of *tnaC* sequence) and 310 nt (observed maximum size). Finally, DNA libraries were prepared for RNA-seq using standard Illumina kits and were run using a NextSeq equipment in a 75 × 2 paired-end scheme.

**Computational processing pipeline of sequencing results**. The computational pipeline used to process the RNA-seq results includes the following steps: (1) performing a quality check on base sequencing quality using fastqc, a program offering analysis on attainment of passing quality scores http://www.bioinformatics. babraham.ac.uk/projects/fastqc, (2) using CUTADAPT[70] to trim adaptor sequences despite low adaptor contamination (<0.5%), (3) creating a reference genome for unique samples of interest. For the gI intron experiments, the reference genome consisted of 30 O-INTERFACE transcripts corresponding to target regions outlined in Fig. 2 (plTetO-asRNA probe-RSE-RBS-ES-RER), 10 O-INTERFACE transcripts corresponding to 5 previously investigated gI intron regions[31] (with and without the adaptor sequence), as well as 2 control O-INTERFACE sequences corresponding to

the sequestered- and free-RBS controls (plTetO-asRNA random asRNA-RSE or mutated RSE, respectively,-RBS-ES-RER). For the sRNA accessibility experiment, the reference genome was comprised of INTERFACE transcripts targeting all sRNA regions in Supplementary Data 2 and 45 regions targeting CsrB and glutamate tRNA (pBAD-asRNA probe-RSE-RBS-ES-RER) (Supplementary Data 7), the sequence of each taRNA and Hfq, as well as sequestered- and free-RBS controls (pBAD-random asRNA-RSE or mutated RSE, respectively,-RBS-ES-RER) as the "reference genome." Step (4) consisted of mapping the RNA-seq reads using BWA MEM for paired-end sequences and, importantly, for the sRNA probing library, excluding all "chimeric" (SA tag) and multi-mapped reads (XA tag). In step (5), resulting sam files were converted to bam using SAMtools[71] and subsequently to a more manageable bed file using BEDTools[72]. The next step (6) exploited awk to develop a script to filter for the R1 reads that contained at least 7 nucleotides of the asRNA probe sequence, and, in the last step, (7), R2 reads corresponding to R1 reads were obtained using their unique identifier (contingent on R1 and R2 reads mapping to the same INTERFACE sequence) and the Linux command "join."

**Calculation of relative accessibility**. A Python code was used to generate a file containing a summary of the number of reads per end position within each unique INTERFACE sequence provided for mapping. The transcript length with respect to each target region was calculated as the number of nucleotides between the observed transcription start site (TSS) for each promoter (consistent with the TSS reported in the literature for pBAD[73] and plTetO[74]) and the transcription end site, both obtained from the RNA-seq results processed following the procedure described above. To calculate relative accessibility, Python was used to calculate the weighted averages of the read length per probe. In the case of experiments probing the gI intron, a baseline (85 nt) was calculated as the minimum O-INTERFACE transcript size and subtracted from each transcript length weighted average. Next, relative accessibility per probe was estimated as the ratio of the adjusted weighted average in the presence of the taRNA (double induction: aTc and ara) to the adjusted weighted average of the transcript length in the absence of any induction. In contrast, for the sRNA experiment, the relative accessibility was calculated utilizing only the weighted average of the transcript length in the presence of the INTERFACE transcript, as the taRNA is natively present. This weighted average of the transcript length was similarly adjusted by a baseline (164 nt) then linearly normalized between 0 and 1 within each sRNA molecule for each replicate to obtain a "raw accessibility" per region and replicate. This was done to discern the hybridization landscape while controlling for transcript abundance and other inconsistencies between strains. Reported accessibilities are average raw accessibility values, linearly normalized within each sRNA for production of figures and tables.

**Proposing Hfq-dependency class from accessibility changes**. The differences in relative accessibility between parent and Hfq-deficient strains were calculated for every region targeted in this study. sRNAs that exhibited any significant differences ($P$-value <0.05, paired-sample two-tailed $t$-test) in normalized accessibility replicates between the two strains were categorized as Hfq-dependent.

**Statistical analyses**. Reported accessibility values were calculated as detailed in "Calculation of relative accessibility." Standard errors (listed in Supplementary Data 2) were calculated for raw accessibilities and propagated through the second normalization ($n = 3$ except when limited by sequencing depth, as indicated in Supplementary Data 2). Accessibility comparisons of sRNA regions between parent and Hfq-deficient strains were performed via paired-sample two-tailed $t$-test (DOF = 2 except when limited by sequencing depth, as indicated in Supplementary Data 2) on normalized accessibility replicates, with stated $P$-values. Comparison of proportions test was performed using an N-1 $\chi^2$-test on all reported accessibilities within "extremes" (>0.875, <0.125) (mapped, DOF = 38, v. unmapped binding sites, DOF = 187) to significance of $P$-value <0.1.

**Proposing novel regulatory regions in sRNA**. Molecular features of regulatory regions within well-characterized sRNAs, specifically (i) location and (ii) average normalized accessibility (>0.75, <0.25) were used as criteria to identify potential regulatory regions in five under-characterized sRNA based on aforementioned observed features in mapped binding sites. Specifically, for the three uncharacterized sRNAs whose predicted mRNA targets were benchmarked, up to three representative likely functional regions were selected for their (i) position in 5'- or 3'-end windows (each encompassing 20% of the total length of the sRNA) and (ii) for "most extreme" high or low accessibility. If regions contained within an sRNA's 5' and 3' windows exhibited mid-accessibility clustering (i.e., no regions stood out in terms of accessibility, e.g., Tpke70), the YUNR RNA:RNA recognition motif[46] was used to narrow the functional region selection. Specifically, of all regions in the 5' half with extreme accessibility (>0.75, <0.25), a representative functional region was chosen as the 5'-most region with containing a significant YUNR motif ($P$-value <0.05). Motif scanning was performed using the FIMO tool within the MEME suite[75].

Due to the accessibility profile of SroH (i.e., no regions in the 3'-end window meeting accessibility extreme criteria listed above), the allowable position range for likely functional region search was extended from 20 to 40% of total sRNA length

from the 5'-end. For the 5'-end of Tpke11 (see Selection of Representative mRNA and sRNA Sequence for EMSAs) that was probed in this work, two regions were believed "likely functional" due to associated absolute transcript lengths within top and bottom ~30% (Tpke11 -1, 3, respectively) of the entire sRNA region library, despite generally poor representation of INTERFACE transcripts targeting other regions in this sRNA.

**Identifying INTERFACE-informed targets from IntaRNA predictions**. Predicted mRNA targets of all sRNAs represented with likely regulatory regions were obtained by IntaRNA[11]. Specifically, the query ncRNA input was the sequence of each represented sRNA (Supplementary Data 1) and viable mRNA target sequence space was delimited as sequences 200 nt upstream and 100 nt downstream of start codons within the annotated NZ_CP009273 genome. Pre-set output and seed parameters were used. Folding window size and maximum basepair distance were set to 150 and 100, respectively, for both target and query RNAs. The top 100 ranked predictions corresponding to each representative sRNA sequence were compiled. Next, predicted interactions were filtered by alignment with proposed regulatory regions. Specifically, predicted mRNAs were only considered viable targets if the sRNA region predicted to bind to the mRNA had at least five overlapping nucleotides with a proposed regulatory region within the sRNA.

For each benchmarked sRNA, the top 5–7 mRNA candidates corresponding to top-ranked or INTERFACE-informed computational prediction group were chosen for experimental validation, if expression profiles permitted (see Selection of Representative mRNA and sRNA Sequence for EMSAs). Importantly, because the top 5 top-ranked and INTERFACE-informed predictions for SroG were identical (due to the identified 3' functional region aligning with >80% of the top 100 computational predictions), we additionally chose to validate all INTERFACE-informed predicted targets that corresponded to the other proposed SroG functional regions (Fig. 4b). For SroA and Tpke11, up to 5 viable targets were tested (see Selection of Representative mRNA and sRNA Sequence for EMSAs). For SroH, the single lowest-energy top-ranked and INTERFACE-informed predictions (given that experimentation was viable, see Selection of Representative mRNA and sRNA Sequence for EMSAs) were tested.

For the subset of sRNAs with highly accessible regions in the 3'-end chosen for biochemical validation (CyaR, GcvB, and GlmY), all viable targets within the top 100 (see Selection of Representative mRNA and sRNA Sequence for EMSAs) adhering to the aforementioned overlap criteria were chosen for experimental verification. Furthermore, to account for strong structure in these terminator regions (presumably the reason for which no CyaR predictions in the top 100 were predicted to utilize the accessible 3' region for interaction), IntaRNA predictions were additionally run for the sequences corresponding to the accessible 3' regions only (starting at the first nucleotide of the accessible sequence and continuing to the 3'-end of the sRNA). Between 1 and 3 functionally interesting targets with the top-10 were ultimately chosen for experimental validation (after viability constraints, see Selection of Representative mRNA and sRNA Sequence for EMSAs). All computational and sequence details of mRNAs tested (and to which likely functional sRNA region they correspond) can be found in Supplementary Data 3.

**Selection of representative mRNA and sRNA sequence for EMSAs**. Experimental logistics of sRNA and mRNA sequences for heterologous expression were aided by transcriptomic data from *E. coli* K-12 MG1655 chromosomally modified to encode a CsrA-FLAG fusion (CML377) in which RNA was extracted in exponential growth phase (OD ~0.6)[47]. Specifically, candidates were deemed unviable if the mRNA expression profile did not include the predicted interaction region of the mRNA, as evaluated by visualizing mapped reads in Integrative Genomics Viewer 2.3 (IGV)[76] (however, candidates without detectable mRNA expression under the aforementioned experimental conditions were not ruled out). Furthermore, candidates were considered unviable if the predicted interaction region within the −200 to +100 mRNA sequence was contained within an upstream coding sequence or if they were annotated as hypothetical proteins in the BW25113 genome.

5'-UTR transcription starts for IVT were chosen to represent those identified by visualizing mapped reads in IGV in an attempt to most accurately mimic in vivo mRNA length and structure. For mRNAs (i) known to be co-transcribed and (ii) showing no obvious transcription start, a pseudo 5'-UTR start was chosen as first nucleotide following the stop codon of the preceding gene. At least 18 nt of the mRNA coding sequence (including start codon) was added following the 5'-UTR to make up the experimentally tested mRNA fragment (Supplementary Data 3). sRNA sequences were chosen based on their reported sequences (Supplementary Data 1) with one exception; in the case of one sRNA selected for experimental validation (Tpke11), the sequence for heterologous expression was extended in order to match previously published Northern blot results (Supplementary Data 3)[44].

**PCR amplification and in vitro transcription (IVT) for EMSA**. PCR amplifications and IVTs for EMSA were performed according to established techniques (see Supplementary Materials and Methods).

**Verification of sRNA:mRNA pairs by EMSA.** Internally labeled sRNAs and respective mRNAs were mixed in 1:10 molar ratios (ranging between 4:40 and 70:700, to normalize for radioactivity signal of sRNAs) in 12–15 μL reactions containing 1X binding buffer (20 mM Tris-HCl at pH 8.0, 1 mM MgCl$_2$, 20 mM KCl, 10 mM Na$_2$HPO$_4$-NaH$_2$PO$_4$ at pH 8.0, 10% glycerol). The reactions were denatured for 5 min at 70 °C, then incubated at 37 °C for 1.5 h. Samples were run for 4 h at 25–35 mA in a 5% non-denaturing polyacrylamide gel with 0.5X TBE running buffer. The gel was dried at 80 °C for 1.5 h (Gel Dryer 583 BioRad) and phospho-imaged using Typhoon FLA 700 (GE Health Life Science).

**Code availability**. All codes used for "Calculation of relative accessibility" can be accessed at https://github.com/mihailom/INTERFACE-pipeline-tools. Scripts pertaining to the SpKG algorithm are available upon request.

## Data availability

Raw RNA-seq data and processed files according to "Computational processing pipeline of sequencing results" section can be accessed at the GEO under accession code GSE117939. All other data are available upon reasonable request.

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

## Acknowledgements

The authors would like to acknowledge Dhivya Arasappan (UT Bioinformatics Consulting Group) for RNA-seq mapping advice, Erika Hale (UT Austin Department of Statistics and Data Sciences) for statistical consultation, Kristofer Reyes (University at Buffalo Department of Materials Design and Innovation) for code consultation, Samuel Stimple (The Ohio State University Department of Chemical and Biomolecular Engineering) for cloning relevant sRNA- and mRNA-coding DNA sequences into plasmids for use in electrophoretic mobility shift assays, as well as Matthew Lab (UT Austin McCombs School of Business), Angela Chen and Runhua Han (UT Austin McKetta Dept. of Chemical Engineering) for their assistance with biochemical assays. We further thank Abigail N. Leistra and Katie Haning (UT Austin McKetta Department of Chemical Engineering) for useful discussions of the manuscript as well as the Texas Advanced Computing Center (TACC) at The University of Texas at Austin for providing high-performance computing resources. We also acknowledge Welch Foundation [F-1756], Air Force Office of Scientific Research Young Investigator program (FA9550-13-1-0160), Consejo Nacional de Ciencia y Tecnología for the graduate fellowship [CONACYT-194638 to J.V.-A.], National Science Foundation (NSF) (CAREER CBET-1254754 to L.M.C.; DGE- 1610403 to M.K.M.).

## Author contributions

Designed research: J.V.-A., M.K.M., Y.L., R.A.L. and L.M.C.; Performed experiments: J. V.-A., P.V., V.F., Y.L. and M.K.M.; Analyzed data: J.V.-A., M.K.M., P.V., V.F., Y.L., R.A.L., and L.M.C.; Wrote paper: M.K.M., J.V.A., P.V., V.F., Y.L., W.B.P., R.A.L. and L. M.C.

## Additional information

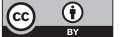

