## [Peer Review File · Nature Communications]

Reviewers' comments:

Reviewer #1 (Remarks to the Author):

In this interesting and innovative paper by Vazquez-Anderson and colleagues the authors describe a novel method, INTERFACE for probing RNA structure in prokaryotes, and in particular identifying sRNA/mRNA pairs that are Hfq and non Hfq dependent. The method relies on transient translations of a library of computationally designed probes that are then read out using high-throughput sequencing. The main claim to novelty is the high throughput nature of the assay, since all of the sRNA:mRNA interactions can be probed simultaneously. Overall the paper is interesting and will certainly be of interest to the prokaryotic community focusing on sRNA:mRNA regulation. I do have several comments the authors will need to address.

1.) It is difficult from the way the paper is written to establish what truly novel insights the method has revealed about sRNA:mRNA interactions. Clearly the Hfq independent binding is interesting, but it is not clear that this is novel. In many ways, overexpression of an anti-sense sRNA with high sequence similarity to the cognate mRNA is not surprising. The authors need to do a better job of saying what truly novel interactions they discovered which were not already predicted by simple RNA folding algorithms.

2.) The authors need some kind of benchmarking to demonstrate that their method identifies interactions that are not computationally predicted.

3.) Along those lines, since the probes are computationally designed, it is not clear that the authors will discover anything unknown since they are by definition only probing perfect anti-sense regions of the genome.

4.) The need to computationally design the library means there are inherent biases and the probing is not truly genome-wide. To estimate these biases the authors must do some form of cross-validation or leave some out estimation. For example if the authors leave out 50% of their probes, how many fewer interactions do they discover. The choice of 700 probes seems somewhat arbitrary and it is important to establish what would happen if they included 1400.

5.) The applicability of INTERFACE is really limited to prokaryotic systems where mRNA:sRNA interactions drive regulation. The authors need to acknowledge this limitation and discuss how such a strategy could work in eukaryotic systems.

Reviewer #2 (Remarks to the Author):

Here are a few comments:

In Fig2c it is curious that complementary regions show high and low accessibility on opposite sides of helices. This is not observed, for example, in SHAPE mapping (Duncan CDS, Weeks KM. SHAPE Analysis of Long-Range Interactions Reveals Extensive and Thermodynamically Preferred Misfolding in a Fragile Group I Intron RNA. *Biochemistry*. 2008;47(33):8504-8513. doi:10.1021/bi800207b.) Deserving of some comment.

In Fig 3, panel b is labeled c.

The p11 claim that over 70% of mapped binding regions are high or low ($0 < x < 0.33$ or $0.66 < x < 1$ for a total of 0.67) does not seem statistically significant on N=192 measurements. At only about one standard deviation, that seems like pretty weak evidence to base other analyses.

On p23.127, Ref 75 seems wrong. Perhaps: DiChiacchio, L., Sloma, M.F. and Mathews, D.H. (2016)

AccessFold: predicting RNA-RNA interactions with consideration for competing self-structure.
Bioinformatics, 32, 1033-1039.

On p24.113 the whole argument of the max function should be enclosed in parentheses to avoid ambiguity.

Response to Reviewers

Reviewer #1 (Remarks to the Author):

In this interesting and innovative paper by Vazquez-Anderson and colleagues the authors describe a novel method, INTERFACE for probing RNA structure in prokaryotes, and in particular identifying sRNA/mRNA pairs that are Hfq and non Hfq dependent. The method relies on transient translations of a library of computationally designed probes that are then read out using high-throughput sequencing. The main claim to novelty is the high throughput nature of the assay, since all of the sRNA:mRNA interactions can be probed simultaneously. Overall the paper is interesting and will certainly be of interest to the prokaryotic community focusing on sRNA:mRNA regulation. I do have several comments the authors will need to address.

1.) It is difficult from the way the paper is written to establish what truly novel insights the method has revealed about sRNA:mRNA interactions. Clearly the Hfq independent binding is interesting, but it is not clear that this is novel. In many ways, overexpression of an anti-sense sRNA with high sequence similarity to the cognate mRNA is not surprising. The authors need to do a better job of saying what truly novel interactions they discovered which were not already predicted by simple RNA folding algorithms.

R: We thank the reviewer for this comment and agree that our results indicating that (i) regional accessibility can be differentially affected by Hfq are not unexpected, as an “RNA structural loosening” effect has been previously proposed in many works. We have adequately acknowledged this throughout the text, ie. in the “Hfq selectively impacts regional sRNA functionality by perturbing accessibility landscapes” section:

“Within this group (169 regions), there was a clear positive skew of hybridization changes in the hfq+ strain relative to the hfq-null mutant (~ 2-fold) (Fig. 6C), supporting the previously proposed structure-relaxing Hfq RNA chaperone mechanism.”

This being addressed, our main intent in this work was not to illustrate a new potential mechanism by which Hfq contributes to sRNA:mRNA interactions. Instead, we wanted to take advantage of the high throughput nature of the INTERFACE approach to jointly interrogate already-characterized Hfq dependency on a large number of sRNAs that has been identified (but not surveyed in *E. coli*). As we describe in the text, using INTERFACE, we are able to propose Hfq hybridization dependency for 14 sRNAs whose Hfq dependence had not previously been characterized in the literature (Supp. Table 5). We also showcase the use of our method for the purposes of identifying regions whose *in vivo* interaction capability depend on distinct intracellular factors (via comparison of +/- factor cellular environments); this analysis was also absent from this literature.

As such, we have included new clarifying statements in the manuscript:

Abstract: “We apply this method with bioinformatics and machine learning approaches to profile over 900 RNA-interaction interfaces in 71 validated, but largely mechanistically under-characterized, sRNAs of Escherichia coli, in the presence and absence of a global regulator, Hfq. Importantly, we showcase the utility of

INTERFACE in detecting RNA regions whose hybridization is Hfq-sensitive, finding two thirds of tested sRNAs with Hfq dependency.”

Introduction: “To demonstrate the ability of *INTERFACE* to capture global impacts of relevant intracellular factors on sRNA accessibility landscapes, we assess sRNA accessibility profiles both in wildtype *E. coli* BW25113 and in an isogenic strain in which the well-characterized Hfq chaperone is knocked out [Baba, 2006 #17]. In this way, we propose hybridization dependency or independence for 14 sRNAs whose relationship with Hfq had not previously been characterized in the literature.”

To address the reviewers concern (ii) regarding truly novel interactions, we clarify both in the manuscript text as well as with additional experimentation as to how we used accessibility data as an *in vivo* filter on computation sRNA target predictions. Briefly, we infer that some sRNA regions are more likely than others to host mRNA binding due to their likelihood of engaging in WC basepairing (as measured by *INTERFACE*). In this way, we obtain *in vivo*-relevant putative targets for biochemical validation assays (more details in R2). To the best of our knowledge, this is the first use of accessibility to inform positive biochemical sRNA target validations, which, in the manuscript, we showcase for six previously uncharacterized sRNAs (SroA, SroE, SroG, SroH, Tpk11, Tpk70).

Furthermore, we have performed additional experiments to follow up on an observation of extreme accessibility in a handful of sRNA terminators. This was especially the case for GlmY, an sRNA that has previously only implicated in protein interaction. By focusing on the terminator region, we found 2 novel mRNA targets, demonstrating the first *in vitro* RNA-RNA interactions involving this sRNA. Furthermore, we conducted EMSAs with truncated regions of the terminator to confirm the importance of the presence of the terminator for target binding. Involvement of sRNA terminators for target binding is still rare in the literature (we discuss only two similar examples that have been published).

We have more clearly conveyed the novelty of this application in the following excerpt (as well as in the re-written section “***Experimental INTERFACE accessibility data inform computational target mRNA prediction***” and the new section “***INTERFACE uncovers novel RNA regulatory potential within the GlmY terminator***”:

Abstract: “Further, we identify *in vivo* hybridization patterns that hallmark functional regions in 16 well-characterized sRNAs and couple these insights to a computational target prediction algorithm to identify novel mRNA targets for six under-characterized and two well-characterized sRNAs. In this way, we biochemically validate twenty-five novel mRNA targets *in vitro*, many of which are not captured by typically-tested, top-ranked computational predictions. In addition, by recognizing extreme 3’ *INTERFACE* accessibility in GlmY, we identify two corresponding novel mRNA targets that rely heavily on the sRNA terminator sequence for interaction.”

2.) The authors need some kind of benchmarking to demonstrate that their method identifies interactions that are not computationally predicted.

R: We thank the reviewer for this insightful critique and attempt to clarify and demonstrate this target-identifying application of INTERFACE. We apologize for the misleading language in regards to the connection between INTERFACE and mRNA target confirmation. Our accessibility probing method is **not a replacement to computational target prediction algorithms**, but rather a method to exploit these useful tools to suggest targets that correspond to regions that exhibit regulatory base-pairing characteristics *in vivo*. Our goal is to use *in vivo* accessibility as a convenient, informative experimental aid to propose computationally-predicted mRNA targets of uncharacterized sRNAs that are most likely to be real *in vivo* (and therefore most worthy of follow-up experimentation). This is motivated by the experimental challenge of using computational approaches alone, in which true positives become considerably sparse after the top ~20 predictions (even in best-performing computational algorithms) and top-ranked targets do not necessarily represent real targets; this has been most recently documented in work by Pain et al., *RNA Biology* 2015, from which the figure below was extracted. Although many computational algorithms are pictured, those pertaining to IntaRNA are the most relevant. We have chosen to utilize this computational method as part of our benchmarking because it is the best performing available algorithm that does not require homology information (CoproRNA, for instance, does). The left panel shows that trusted sRNA targets predicted by IntaRNA have been seen to rank anywhere in the top hundreds of thermodynamic predictions. The right panel displays the drastic drop in true positive frequency after ~ top 20 predictions. These two limitations of computational prediction algorithms establish a need for the use of *in vivo* data to help determine most likely true targets that can be further validated experimentally. Traditionally, this *in vivo* filtering has been limited to low throughput co-immunoprecipitation studies (that are much more labor intensive than INTERFACE) or to high throughput studies that are not suitable for all sRNAs (see below). Here we propose the collection of high throughput *in vivo* accessibility data by INTERFACE, to yield “INTERFACE-informed computationally predicted targets” from large collections of computational (IntaRNA) predictions that are calculated in the absence of any *in vivo* results.

Figure from Pain, et al., *RNA Biology*, (2015)

We have clarified this concept in the completely-overhauled section previously titled “INTERFACE functional region recognition informs target mRNA prediction” (now titled “Experimental INTERFACE accessibility data

inform computational target mRNA prediction”). One excerpt that particularly pertains to the limitations of computational prediction algorithms can be seen:

“We next hypothesized that experimentally-determined regional accessibility by INTERFACE could be coupled to computational sRNA:mRNA predictions to identify mRNA targets that are most relevant to *in vivo* functionality. This approach is motivated by the inability for even the best-performing computational thermodynamic prediction algorithms, such as CopraRNA and IntaRNA,¹⁸ to account for the intracellular environment; this shortcoming often leads to low ranking of true targets (below hundreds of predicted pairings) as well as high rankings (i.e., those within the top 10 predictions) that do not correspond to any experimentally confirmed targets^{18,59}. To aid and improve identification of most-likely true targets, we propose a pipeline that begins with the identification of likely functional RNA regions from INTERFACE accessibility data (Fig. 4A), based on extreme (high or low) accessibility and location in the RNA. These functional regions can then be used as guides to filter results of computational sRNA target predictions, to obtain a reduced list of computational targets (Fig. 4B, left). Importantly, the filtered list includes targets of any computed energy rank, as long as their predicted sRNA binding region was found to exhibit extreme accessibility *in vivo* by INTERFACE (Fig. 4B, right).”

We have further re-designed Figure 4 to visually clarify the two steps of the pipeline: the use of INTERFACE data to identify likely functional sRNA regions in uncharacterized sRNA (A) and filtering out computational predictions to consider only those that rely on the proposed *in vivo* functional regions for interaction (B).

We also appreciate and agree with the reviewer’s recommendation for benchmarking our method to strengthen the manuscript; to this end, we have compared the positive predictive value (PPV) of “INTERFACE-informed” targets against typically-tested, top-ranked computationally-only predicted targets (Fig. 4B, right) (with no prior *in vivo* information concerning binding partners). In this sense, this group represents sRNAs whose low expression (as it is typical for many sRNAs) or independence from RNA chaperone proteins limits their target network characterization by more direct high throughput methods such as RIP-seq (Melamed, et al., *Mol. Cell*, 2016)/CLASH methods (Waters, et al., *The EMBO Journal* 2017; Liu, et al., *BMC Genomics* 2017).

In this new manuscript section and newly revised corresponding Figure 4, we outline the use of INTERFACE-gathered *in vivo* accessibility data for the selection of likely functional regions for three sRNAs that are used in our benchmarking study (Fig. 4C). We also comment on the PPV of each predictive group (INTERFACE-informed and top-ranked computational predictions) for each sRNA tested (Fig. 4D), including rank and identity of positive mRNA targets (Fig. 4E). Importantly, to adequately benchmark the value of using accessibility data for the identification of true targets from computational predictions, we performed a total of 25 additional *in vitro* binding assays to benchmark INTERFACE-informed predictions against typically-tested, top-ranked targets (no prior *in vivo* information) for 3 sRNAs. We now present this analysis and confirm that coupling computational target predictions with accessibility data offers a pool of true mRNA targets that are typically not considered for experiments based on them being computationally lowly-ranked (as low as 76/100 for sRNA SroG) in conventional approaches that focus on selecting top-ranked computationally predicted for experimental

verification. This is notable considering that computational prediction accuracy falls off drastically after ~ 20/100, as previously mentioned; as such, most sRNA target characterization works in the literature do not test lowly ranked targets in the absence of large amounts of experimental data (i.e. low throughput, labor intensive co-immunoprecipitations that are typically done for one sRNA at a time).

As an additional attempt to address this concern, we explicitly point out GlmY as an example in which computational predictions altogether failed to predict one of our confirmed targets in the top 100 predictions in the absence of being guided by INTERFACE data—we became aware of this potential target only upon constraining predictions to the highly accessible terminator sequence.

3.) Along those lines, since the probes are computationally designed, it is not clear that the authors will discover anything unknown since they are by definition only probing perfect anti-sense regions of the genome.

R. We thank the reviewer for this observation. Although our probes are perfect asRNAs, they exhibit widely different binding activities *in vivo* (supported statistically, Supp. Table 2), likely due to RNA secondary structure and/or RNA interaction with native intracellular factors. The reviewer does bring up a good point concerning our understanding of trans sRNA:mRNA interactions to date; more specifically, that these interactions contain imperfect complementarity and gaps. However, to account for the sample space of all possible imperfect complementarity would be computationally and experimentally expensive. Instead, we purposefully computationally designed target regions to (i) support full hybridization coverage of all sRNAs as well as (ii) to capture regions most likely to exhibit high activity (as predicted by the biophysical model within the machine learning algorithm), which, on average, is approximately 1 region for every 10 nucleotides.

4.) The need to computationally design a the library means there are inherent biases and the probing is not truly genome-wide. To estimate these biases the authors must do some form of cross-validation or leave some out estimation. For example if the authors leave out 50% of their probes, how many fewer interactions do they discover. The choice of 700 probes seems somewhat arbitrary and it is important to establish what would happen if they included 1400.

R. The reviewer is absolutely correct that our probing approach is not genome wide; rather, the probing is user-defined (and requires the user to be aware of the sequence of the target RNA is known). As far as inherent biases due to the regions chosen to be targeted, target regions were purposefully designed for full coverage of each target sRNA. Given that target regions were chosen to minimize the number of asRNA probing strains, if we were to exclude even a small fraction of probes, we would lose full coverage of the probing. In this case the experiments would not yield meaningful data as the probed accessibility landscape for each sRNA is calculated relative to the other target regions probed within the sRNA. However, the reviewer brings up a great point that, the more regions that are probed along a sRNA, the more information is gained concerning the binding accessibility profile. To this end, the user is more likely to sense “true” binding sites by increasing the number

of probes covering an uncharacterized sRNA (and we have seen interesting differences in the well-characterized sRNAs in which we purposefully probed mapped binding sites in addition to those chosen by the machine learning algorithm, see below). In this research; however, we have experimentally determined that an average coverage of 1 region per 10 nucleotides is sufficient to obtain comprehensive data, while minimizing the cloning needed for constructing larger probing libraries. We have now clearly pointed out these trade-offs in the discussion:

“In this work, we detail the design and validation of INTERFACE, a synthetic transcription elongation-based reporter system that supports simultaneous in vivo sensing of accessible interfaces within an ensemble of user-defined RNA regions via RNA-seq.”

“INTERFACE may also offer insights into sRNA:mRNA binding initiation, as supported by its sensitivity to very subtle shifts in the target region window. Specifically, regions contained within previously-mapped mRNA-binding sites sometimes have vastly different accessibilities than regions with slightly different indices..... These observations speak to the importance of probing the full length of mapped binding sites, or, if such sites are unknown, increasing the sample space by interrogating overlapping or partially redundant stretches with representative lengths (e.g., the average length of known sRNA:mRNA interaction sites in the organism of interest, or guided by free energy considerations).”

5.) The applicability of INTERFACE is really limited to prokaryotic systems where mRNA:sRNA interactions drive regulation. The authors need to acknowledge this limitation and discuss how such a strategy could work in eukaryotic systems.

R. The authors acknowledge that the current system is limited strictly to prokaryotic systems due to its exploitation of transcription-translation coupling. However, because the premise of the targeting and reporter signaling within this system is based on formation and disruption of RNA WC basepairing, it can theoretically be applied to any domain of life as long as the antitermination (AT) mechanism is replaced accordingly. As eukaryotic RNA polymerases have been shown sensitive to hairpin terminators *in vitro* (Komissarova, et al., *Mol. Cell* 2002 and Nielsen, et al., *Science* 2013), an adaptation in which the stability of the hairpin structure is determined by the extent of asRNA binding to its target region may be a viable for use in eukaryotes. In this way, the adaptation would be similar to the RBS exposure determination by asRNA binding in the current system. We have proposed this briefly in the discussion :

“More broadly, given that INTERFACE mostly exploits formation and disruption of of Watson-Crick basepairing, we foresee that INTERFACE will support in vivo investigation of RNA function in other organisms pending appropriate replacement of the antitermination (AT) mechanism. Importantly, TnaC peptides have been identified in species of many pathogenically-relevant bacterial genera (e.g. Vibrio, Yersinia, Shigella)⁷⁷, suggesting that they share similar ribosomal stalling-based AT mechanisms with E. coli. Furthermore, as eukaryotic RNA polymerases have been shown sensitive to hairpin terminators in vitro^{78, 79}, an adaptation in which the stability of the hairpin structure is determined by the extent of asRNA binding to its target region could be viable for use in higher organisms.”

However, the application of this novel method to eukaryotic systems is beyond the scope of the work presented.

Reviewer #2 (Remarks to the Author):

In Fig2c it is curious that complementary regions show high and low accessibility on opposite sides of helices. This is not observed, for example, in SHAPE mapping (Duncan CDS, Weeks KM. SHAPE Analysis of Long-Range Interactions Reveals Extensive and Thermodynamically Preferred Misfolding in a Fragile Group I Intron RNA. Biochemistry. 2008;47(33):8504-8513. doi:10.1021/bi800207b.) Deserving of some comment.

R. The reviewer brings up an important point that accessibility profiles are often very different from SHAPE reactivity profiles and an interesting observation that complementary sides of stem loops often have varying accessibilities. First, we would like to mention that differences in results between these methods have been previously reported, as we include in the introduction:

“As these local nucleotide availabilities do not always correlate with regional-level accessibility that more accurately mimics RNA:RNA interactions of regulatory interfaces⁴¹, efforts have also been placed on methods to quantify regional in vivo RNA hybridization.”

It is not surprising that complementary sides of helices have varying accessibilities because we anticipate that the ability for asRNA hybridization to access a target region would be much more easily disrupted by the molecular arrangement of the target RNA than modification abilities of a small chemical molecule. For example, chemical probing methods are clearly sensitive to basepairing but are likely less sensitive to spatial occlusions of certain sides of helices (near long-range contacts, etc.) than asRNAs, merely due to size difference. We now include this point in the “Validating molecular features that enable INTERFACE to capture regional accessibility” section:

“It is interesting to note that, in contrast to chemical footprinting data typical of gI introns⁵⁴, complementary sides of stem-loops can exhibit accessibility differences (cf. region 9 and 13). Although discrepancies between accessibility and footprinting methods have been previously recognized⁴⁶, this observation supports a heightened sensitivity of accessibility probes to spatial arrangement of the taRNA.”

We would like to point out, however, that the gI intron in the suggested publication is also from a different organism than the one showcased in the paper so a direct comparison is difficult to make.

In Fig 3, panel b is labeled c.

R. We thank the reviewer for pointing out this oversight, and have fixed it.

The p11 claim that over 70% of mapped binding regions are high or low ($0 < x < 0.33$ or $0.66 < x < 1$ for a total of

0.67) does not seem statistically significant on N=192 measurements. At only about one standard deviation, that seems like pretty weak evidence to base other analyses.

R. We agree with the reviewer that 70% does not seem particularly strong evidence off of which to justify further analyses. We rather justify the downstream use of this characteristic based on a comparison [of proportions] test between proportion of mapped versus unmapped binding sites exhibiting extreme accessibility (and find that mapped binding sites have a higher proportion of regions that fall into this criteria to P-value < 0.1). This test gives a sense for “enrichment” of extreme accessibilities in one group over the other. We have adjusted the text to read:

“Upon performing and analyzing results of the INTERFACE experiment, a comparison of proportions test confirmed that exact binding regions (38) (Fig. 3B) exhibited a significantly increased proportion in top 25% accessibility “extremities” (less than 0.125 or greater than 0.875 on a 0 to 1 scale) compared to the regions chosen for INTERFACE probing of these same sRNAs by the machine learning algorithm (179 regions) (P-value < 0.1) (Fig. 3B).”

On p23.127, Ref 75 seems wrong. Perhaps: DiChiacchio, L., Sloma, M.F. and Mathews, D.H. (2016) AccessFold: predicting RNA-RNA interactions with consideration for competing self-structure. Bioinformatics, 32, 1033-1039.

R. We thank the reviewer for pointing out this mistake and have addressed it.

On p24.113 the whole argument of the max function should be enclosed in parentheses to avoid ambiguity.

R. We thank the reviewer for pointing out this ambiguity, and have since clarified.

REVIEWERS' COMMENTS:

Reviewer #1 (Remarks to the Author):

The authors have addressed all of my concerns and the paper can be published.